# CellSAM: a foundation model for cell segmentation

Markus Marks [1,2,6], Uriah Israel[1,3,6], Rohit Dilip[1], Qilin Li[2], Changhua Yu[3], Emily Laubscher [4], Ahamed Iqbal[3], Elora Pradhan[3], Ada Ates[3], Martin Abt [3], Caitlin Brown[3], Edward Pao[3], Shenyi Li [3], Alexander Pearson-Goulart[3], Pietro Perona[1,2], Georgia Gkioxari[1], Ross Barnowski [3], Yisong Yue [1] & David Van Valen [3,5] ✉

Cells are a fundamental unit of biological organization, and identifying them in imaging data—cell segmentation—is a critical task for various cellular imaging experiments. Although deep learning methods have led to substantial progress on this problem, most models are specialist models that work well for specific domains but cannot be applied across domains or scale well with large amounts of data. Here we present CellSAM, a universal model for cell segmentation that generalizes across diverse cellular imaging data. CellSAM builds on top of the Segment Anything Model (SAM) by developing a prompt engineering approach for mask generation. We train an object detector, CellFinder, to automatically detect cells and prompt SAM to generate segmentations. We show that this approach allows a single model to achieve human-level performance for segmenting images of mammalian cells, yeast and bacteria collected across various imaging modalities. We show that CellSAM has strong zero-shot performance and can be improved with a few examples via few-shot learning. Additionally, we demonstrate how CellSAM can be applied across diverse bioimage analysis workflows. A deployed version of CellSAM is available at https://cellsam.deepcell.org/.

Accurate cell segmentation is crucial for quantitative analysis and interpretation of various cellular imaging experiments. Modern spatial genomics assays can produce data on the location and abundance of $10^2$ protein species and $10^3$ RNA species simultaneously in living and fixed tissues[1–5]. Accurate cell segmentation allows this type of data to be converted into interpretable tissue maps of protein localization and transcript abundances; these maps provide important insights into the biology of healthy and diseased tissues. Similarly, live-cell imaging provides insight into dynamic phenomena in bacterial and mammalian cell biology. Studying live-cell imaging data has provided mechanistic insights into critical phenomena such as the mechanical behavior of the bacterial cell wall[6,7], information transmission in cell signaling pathways[8–13], heterogeneity in immune cell behavior during immunotherapy[14] and the morphodynamics of development[15]. Cell segmentation is also a key challenge for these experiments, as cells must be segmented and tracked to create temporally consistent records of cell behavior that can be queried at scale. These methods have seen use in several systems, including mammalian cells in cell culture[13,16] and tissues[5], bacterial cells[17–20] and yeast[21–23].

Considerable progress has been made in recent years on the problem of cell segmentation, driven primarily by advances in deep learning[24]. Progress in this space has occurred mainly in two distinct directions. The first direction seeks to find deep learning architectures that achieve state-of-the-art performance on cellular imaging

[1]Division of Computing and Mathematical Sciences, Caltech, Pasadena, CA, USA. [2]Division of Engineering and Applied Science, Caltech, Pasadena, CA, USA. [3]Division of Biology and Biological Engineering, Caltech, Pasadena, CA, USA. [4]Division of Chemistry and Chemical Engineering, Caltech, Pasadena, CA, USA. [5]Howard Hughes Medical Institute, Chevy Chase, MD, USA. [6]These authors contributed equally: Markus Marks, Uriah Israel. ✉e-mail: vanvalen@caltech.edu

tasks. These methods have historically focused on a particular imaging modality (for example, brightfield imaging) or target (for example, mammalian tissue) and have difficulty generalizing beyond their intended domain[25–31]. For example, Mesmer's[28] representation for a cell (cell centroid and boundary) enables good performance in tissue images but would be a poor choice for elongated bacterial cells. Similar tradeoffs in representations exist for the current collection of Cellpose models, necessitating the creation of a model zoo[26,32]. The second direction is to work on improving labeling methodology. Cell segmentation is an application of the instance segmentation problem, which requires pixel-level labels for every object in an image. Creating these labels can be expensive (US$0.01 per label, with hundreds to thousands of labels per image)[28,33], which provides an incentive to reduce the marginal cost of labeling. A recent improvement to labeling methodology has been human-in-the-loop labeling, where labelers correct model errors rather than produce labels from scratch[26,28,34]. Further reductions in labeling costs can increase the amount of labeled imaging data by orders of magnitude.

Recent work in machine learning on foundation models holds promise for providing a complete solution. Foundation models are large deep neural network models (typically transformers[35]) trained on large amounts of data in a self-supervised fashion with supervised fine-tuning on one or several tasks[36]. Foundation models include the GPT[37,38] family of models, which have proven transformative for natural language processing[36]. These types of attention-based models have recently been used for processing biological sequences[39–43]. These successes have inspired similar efforts in computer vision. The Vision Transformer (ViT)[44] was introduced in 2020 and has since been used as the basis architecture for a collection of vision foundation models[45–49]. A key feature of foundation models is the scaling of model performance with model size, dataset size and compute[50]; these scaling laws have been observed for both language and vision models[51]. These scaling laws offer a path toward generalist models for cellular image analysis by increasing dataset and model size in exchange for dealing with the increased compute cost of training foundation models. This is in contrast to previous efforts that have focused on model architecture design and representation engineering.

One recent foundation model well suited to cellular image analysis is the Segment Anything Model (SAM)[52]. This model uses a ViT to extract information-rich features from raw images. These features are then directed to a module that generates instance masks based on user-provided prompts, which can be either spatial (for example, an object centroid or bounding box) or semantic (for example, an object's visual description). Notably, the promptable nature of SAM enabled scalable dataset construction, as preliminary versions of SAM allowed labelers to generate accurate instance masks with 1–2 clicks. The final version of SAM was trained on a dataset of 11 million images containing over 1 billion masks and demonstrated strong performance on various zero-shot evaluation tasks. Recent work has attempted to apply SAM to problems in biological and medical imaging, including medical image segmentation[53–56], lesion detection in dermatological images[57,58], nuclear segmentation in hematoxylin and eosin (H&E) images[59,60] and cellular image data for use in the Napari software package[61].

Works such as MicroSAM[61] or MedSAM[56] use SAM's original workflow to speed up annotation of cells and medical data, label a large dataset and then fine-tune the original SAM model. However, reliable automated segmentation is still missing in these works. Although promising, these studies reported challenges adapting SAM to these new use cases[53,61]. These challenges include reduced performance and uncertain boundaries when transitioning from natural to medical images. Cellular images contain additional complications: they can involve different imaging modalities (for example, phase microscopy versus fluorescence microscopy), thousands of objects in a field of view (FOV) (as opposed to dozens in a natural image) and uncertain and noisy boundaries (artifacts of projecting three-dimensional objects into a two-dimensional plane)[61].

In addition to these challenges, SAM's default strategy for automatic prompting does not allow for accurate inference on cellular images. SAM's automated prompting uses a uniform grid of points to generate masks, an approach that is poorly suited to cellular images given the wide variation of cell densities. More precise prompting (for example, a bounding box or mask) requires prior knowledge of cell locations. Because cellular images often contain a large number of cells, it is impractical for users to provide prompts to SAM manually. This limitation makes it challenging for SAM to serve as a foundation model for cell segmentation because it still requires substantial human input for inference. A solution that enables the automatic generation of prompts would enable SAM-like models to serve as foundation models and knowledge engines, as they could accelerate the generation of labeled data, learn from them and make that knowledge accessible to life scientists via inference.

In this work, we developed CellSAM, a foundation model for cell segmentation (Fig. 1). CellSAM extends the SAM methodology to perform automated cellular instance segmentation. To achieve this, we first assembled a comprehensive dataset for cell segmentation spanning five broad data archetypes: tissue, cell culture, yeast, H&E and bacteria. Critically, we removed data leaks between training and testing data splits to ensure an accurate assessment of model performance. To automate inference with CellSAM, we developed CellFinder, a transformer-based object detector that uses the Anchor DETR framework[62]. CellSAM and CellFinder share SAM's ViT backbone for feature extraction; these features are first used by CellFinder to generate bounding boxes around the cells to be used as prompts for SAM. The bounding boxes (prompts) and ViT features are fed into a decoder to generate instance segmentations of the cells in an image. We trained CellSAM on a large, diverse corpus of cellular imaging data, enabling it to achieve state-of-the-art performance across 10 datasets. We also evaluated CellSAM's zero-shot performance using a held-out dataset, LIVECell[63], demonstrating that it substantially outperforms existing methods for zero-shot segmentation. A deployed version of CellSAM is available at https://cellsam.deepcell.org.

## Results

### Construction of a dataset for a generalist cell segmentation model

A major challenge with existing cellular segmentation methods is their inability to generalize across cellular targets, imaging modalities and cell morphologies. To address this, we curated a dataset from the literature containing two-dimensional images from a diverse range of targets (mammalian cells in tissues and adherent cell culture, yeast cells, bacterial cells and mammalian cell nuclei) and imaging modalities (fluorescence, brightfield, phase contrast and mass cytometry imaging).

Our final dataset consisted of TissueNet[28], DeepBacs[64], BriFiSeg[65], Cellpose[25,26], Omnipose[66,67], YeastNet[68], YeaZ[69], the 2018 Kaggle Data Science Bowl (DSB) dataset[70], a collection of H&E datasets[71–77] and an internally collected dataset of phase microscopy images across eight mammalian cell lines (Phase400). We group these datasets into six types for evaluation: Tissue, Cell Culture, H&E, Bacteria and Yeast. As the DSB[70] comprises cell nuclei that span several of these types, we evaluate it separately and refer to it as Nuclear, making a total of six categories for evaluation. Although our method focuses on whole-cell segmentation, we included DSB[70] because cell nuclei are often used as a surrogate when the information necessary for whole-cell segmentation (for example, cell membrane markers) is absent from an image. Figure 2a shows the number of annotations per evaluation type. Finally, we used a held-out dataset, LIVECell[63], to evaluate CellSAM's zero-shot performance. This dataset was curated to remove low-quality images and images that did not contain sufficient information about the boundaries of closely packed cells. A detailed

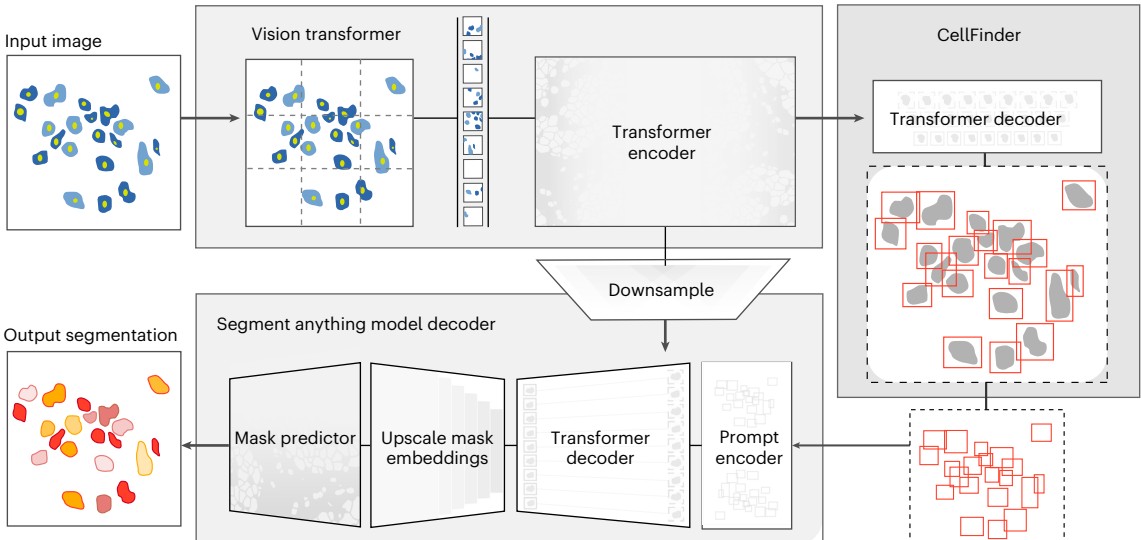

**Fig. 1 | CellSAM: a foundational model for cell segmentation.** CellSAM combines SAM's mask generation and labeling capabilities with an object detection model to achieve automated inference. Input images are divided into regularly sampled patches and passed through a transformer encoder (that is, a ViT) to generate information-rich image features. These image features are then sent to two downstream modules. The first module, CellFinder, decodes these features into bounding boxes using a transformer-based encoder–decoder pair. The second module combines these image features with prompts to generate masks using SAM's mask decoder. CellSAM integrates these two modules using the bounding boxes generated by CellFinder as prompts for SAM. CellSAM is trained in two stages, using the pretrained SAM model weights as a starting point. In the first stage, we train the ViT and the CellFinder model together on the object detection task. This yields an accurate CellFinder but results in a distribution shift between the ViT and SAM's mask decoder. The second stage closes this gap by fixing the ViT and SAM mask decoder weights and fine-tuning the remainder of the SAM model (that is, the model neck) using ground truth bounding boxes and segmentation labels.

description of data sources and preprocessing steps can be found in Appendix A. Our full, preprocessed dataset is publicly available at https://cellsam.deepcell.org.

## CellSAM creates masks using box prompts generated from CellFinder

In early experiments, we found that providing ground truth bounding boxes as prompts to SAM (ground truth prompts represent an upper bound on performance) achieved substantially higher zero-shot performance than point prompting (Extended Data Fig. 1). This is in agreement with previous analyses of SAM applied to biological[61] and medical[53] images. Because the ground truth bounding box prompts yield accurate segmentation masks from SAM across various datasets, we sought to develop an object detector that could generate prompts for SAM in an automated fashion. Given that our zero-shot experiments demonstrated that ViT features can form robust internal representations of cellular images, we reasoned that we could build an object detector using the image features generated by SAM's ViT. Previous work explored this space and demonstrated that ViT backbones can achieve state-of-the-art performance on natural images[78,79]. For our object detection module, we use the Anchor DETR[62] framework with the same ViT backbone as the SAM module; we call this object detection module CellFinder. Anchor DETR is well suited for object detection in cellular images because it formulates object detection as a set prediction task. This allows it to perform cell segmentation in images with densely packed objects, a common occurrence in cellular imaging data. Alternative bounding box detection methods (for example, the R-CNN family) rely on non-maximum suppression (NMS)[80,81], leading to poor performance in this regime. Methods that frame cell segmentation as a dense, pixel-wise prediction task (for example, Mesmer[28], Cellpose[25] and Hover-net[30]) assume that each pixel can be uniquely assigned to a single cell and cannot handle overlapping objects.

The ground truth prompting scheme by itself does not achieve real-world performance standards. Our analysis showed that SAM cannot accurately segment many cell types, likely due to the distribution of images seen during training. To adapt CellSAM from natural images to cellular images, we fine-tune the SAM model neck (the layers connecting SAM's ViT to its decoder) while leaving other layers frozen to retain generalization ability. Training CellSAM in this manner achieved state-of-the-art accuracy when provided with ground truth bounding box prompts (Supplementary Fig. 1).

We train CellSAM in two stages; the full details can be found in the supplementary materials. In the first stage, we train CellFinder on the object detection task. We convert the ground truth cell masks into bounding boxes and train the ViT backbone and the CellFinder module. Once CellFinder is trained, we freeze the model weights of the ViT and fine-tune the SAM module as described above. This accounts for the distribution shifts in the ViT features that occur during the CellFinder training. Once training is complete, we use CellFinder to prompt SAM's mask decoder. We refer to the collective method as CellSAM; Fig. 1 outlines an image's full path through CellSAM during inference.

## Benchmarking CellSAM's performance on numerous biological datasets

We benchmarked CellSAM's performance using F1 error (1 − F1) as metric (Fig. 2b) against Cellpose, a widely used cell segmentation algorithm. Because our work includes both dataset and model development, we chose benchmarks that allow us to measure the contributions of data and model architecture to overall performance. Our benchmarks include comparisons to a pretrained generalist Cellpose model (cyto3), an internally trained generalist Cellpose model and a suite of internally trained specialist (that is, trained on a single dataset) Cellpose models. Internally trained models were trained on the CellSAM dataset or a suitable subset using previously published training recipes, whereas evaluations were performed on a held-out split of the same dataset. We further evaluated CellSAM's performance on the evaluation split of the NeurIPS Cell Segmentation Challenge[82] (Fig. 2b). For this evaluation, we fine-tuned CellSAM with an additional hematology dataset, which was

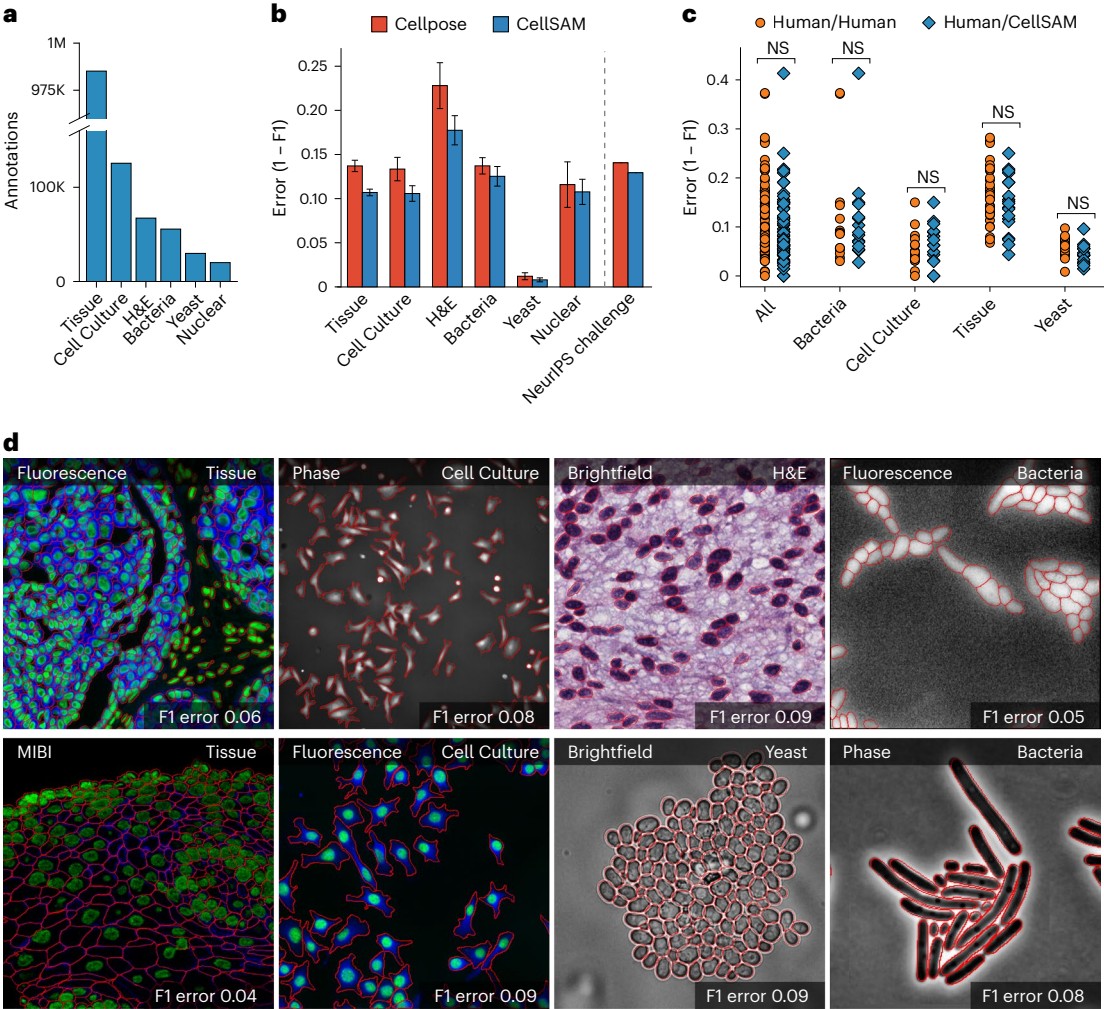

**Fig. 2 | CellSAM is a strong generalist model for cell segmentation. a**, For training and evaluating CellSAM, we curated a diverse cell segmentation dataset from the literature. The number of annotated cells is given for each data type. Nuclear refers to a heterogeneous dataset (DSB)[70] containing nuclear segmentation labels. **b**, Segmentation performance for CellSAM and Cellpose across different data types. We compared the segmentation error (1 − F1) for models that were trained as generalists (that is, the full dataset). Models were trained for a similar number of steps across all datasets. We observed that CellSAM-generalist had a lower error than Cellpose-generalist on all tested data categories. Furthermore, we validated this finding on a held-out competition dataset from the Weakly Supervised Cell Segmentation in Multi-modality

High-Resolution Microscopy Images (that is, the NeurIPS challenge)[91]. Error bars were computed by computing the segmentation error per image and then calculating the mean and s.e. The categories contained the following number of samples: Tissue = 330, Cell Culture = 144, H&E = 51, Bacteria = 260, Yeast = 32 and Nuclear = 56. **c**, Human versus human and CellSAM-generalist versus human (CellSAM/human) inter-rater performance comparison. A two-sided *t*-test confirms that no statistical difference exists between CellSAM and human performance. **d**, Qualitative results of CellSAM segmentations for different data and imaging modalities. Predicted segmentations are outlined in red. NS, not significant.

a substantial fraction of the NeurIPS challenge dataset. In almost every comparison, we found that CellSAM outperformed generalist Cellpose models (whether pretrained or internally trained) and was equivalent to specialist Cellpose models trained exclusively on individual datasets. We highlight features of our benchmarking analyses below.

- **CellSAM is a strong generalist model**. Generalization across cell morphologies and imaging datasets has been a major challenge for deep-learning-based cell segmentation algorithms. To evaluate CellSAM's generalization capabilities, we compared the performance of CellSAM and Cellpose models trained as specialists (that is, on a single dataset) to generalists (that is, on all datasets). Consistent with the literature, we observe that Cellpose's performance degraded when trained as a generalist (Extended Data Fig. 3). By contrast, we found that the performance of CellSAM-generalist was equivalent to or better than CellSAM-specialist across all

data categories and datasets (Extended Data Fig. 3). Moreover, CellSAM-generalist outperformed Cellpose-generalist in all data categories (Fig. 2b and Extended Data Figs. 2 and 3). This analysis highlights an essential feature of a foundational model: maintaining performance with increasing data diversity and scale.

- **CellSAM achieves human-level accuracy for generalized cell segmentation**. We use the error (1 − F1) to assess the consistency of segmentation predictions and annotator masks across a series of images. We compared the annotations of three experts with each other (human versus human) and with CellSAM (human versus CellSAM). This comparison explores whether CellSAM's performance is within the error margin created by annotator preferences (for example, the thickness of a cell boundary). We compared annotations across four data categories: mammalian cells in tissue, mammalian cells in cell culture, bacterial cells and yeast cells. A two-sided *t*-test revealed no significant differences

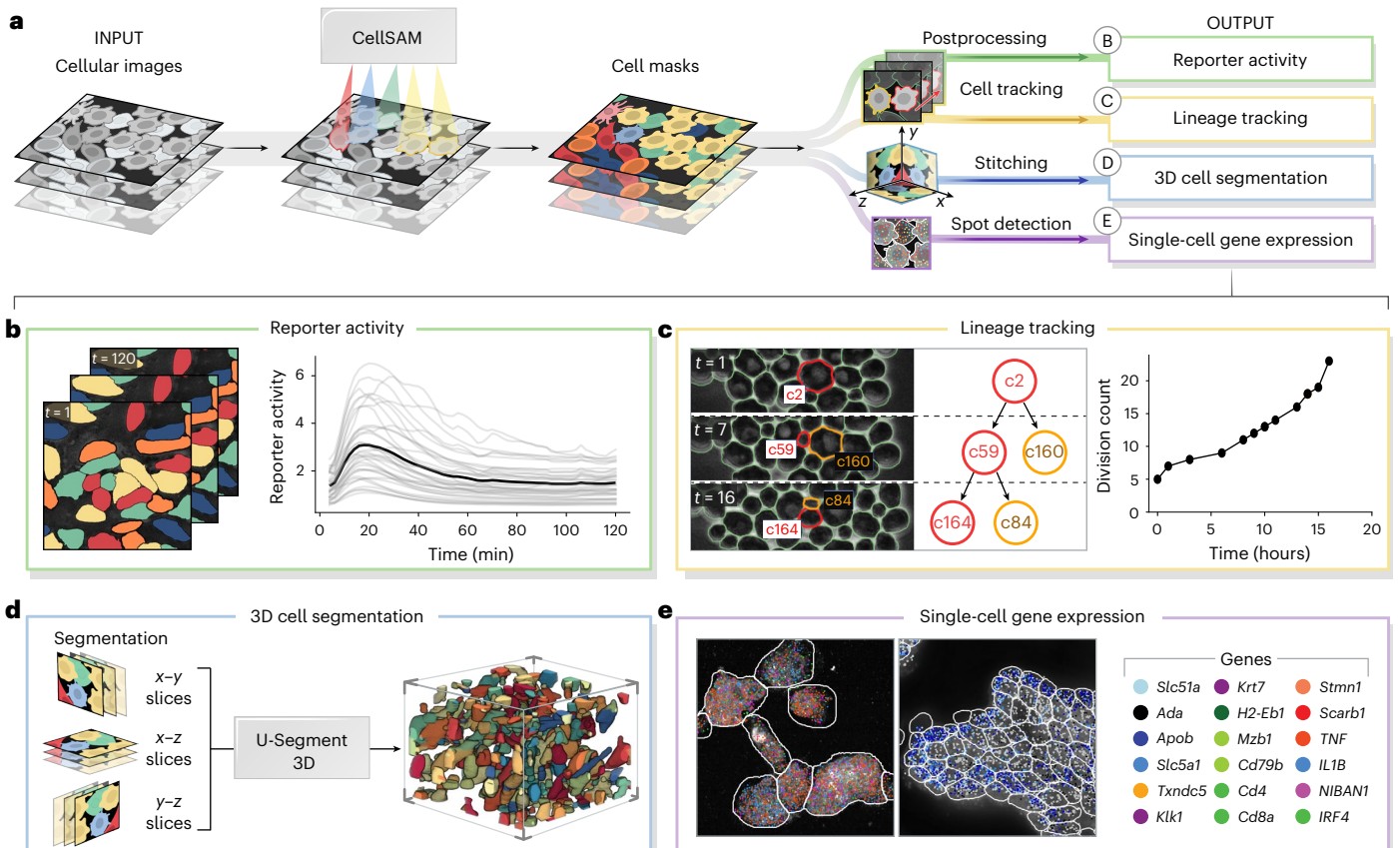

**Fig. 3 | CellSAM enables diverse bioimage analysis workflows.** Because CellSAM-generalist functions across image modalities and cellular targets, it can be immediately applied across bioimaging analysis workflows without requiring task-specific adaptations. **a**, We schematically depict how CellSAM-generalist fits into the analysis pipeline for live-cell imaging and spatial transcriptomics, eliminating the need for different segmentation tools and expanding the scope of possible assays to which these tools can be applied. **b**, Segmentations from CellSAM are used to track cells[87] and quantify fluorescent live-cell reporter activity in cell culture. **c**, CellSAM segments cells in multiple frames from a video of budding yeast cells. These cells are tracked across frames using a tracking algorithm[87] that ensures consistent identities, enabling accurate lineage construction and cell division quantification. **d**, CellSAM is used to segment slices of a three-dimensional image, and these segmented slices are fed into u-Segment3D[89] to create a three-dimensional segmentation. **e**, Segmentations generated using CellSAM are integrated with Polaris[85], a spatial transcriptomics analysis pipeline. Because of CellSAM's generalist nature, we can apply this workflow across sample types (for example, tissue and cell culture) and imaging modalities (for example, seqFISH and MERFISH). Datasets of cultured macrophage cells (seqFISH) and mouse ileum tissue (MERFISH)[86] were used to generate the data in this example. MERFISH segmentations were generated with CellSAM with an image of a nuclear and membrane stain; seqFISH segmentations were generated with CellSAM with a maximum intensity projection image of all spots. 3D, three-dimensional.

between these two comparisons, indicating that CellSAM's outputs are similar to expert human annotators (Fig. 2c). This is demonstrated by non-significant $P$ values between CellSAM-annotator and inter-annotator agreements, specifically for Tissue: $P = 0.18$, Cell Culture: $P = 0.49$, Yeast: $P = 0.11$ and Bacteria: $P = 0.90$.

- **CellSAM enables fast and accurate labeling**. When provided with ground truth bounding boxes, CellSAM achieves high-quality cell masks without any fine-tuning on unseen datasets (Extended Data Fig. 1). Because drawing bounding boxes consumes considerably less time than drawing individual masks, this means that CellSAM can be used to generate highly accurate labels quickly, even for out-of-distribution data.
- **CellSAM is a strong zero-shot and few-shot learner**. We used the LIVECell dataset to explore CellSAM's performance in zero-shot and few-shot settings. We stratified CellSAM's zero-shot by cell lines present in LIVECell (Extended Data Figs. 4b and 5). We found that although performance varied by cell line, we could recover adequate performance in the few-shot regime for a number of the cell lines (for example, A172). Extended Data Fig. 4 shows that CellSAM improves its performance with only 10 additional FOVs ($10^2$–$10^3$ cells) for each cell line. We found that fine-tuning could

not recover performance for cell lines with morphologies far from the training data distribution (for example, SH-SY5Y). This may reflect a limitation of bounding boxes as a prompting strategy for SAM models.

## CellSAM enables diverse bioimage analysis workflows

Cell segmentation is a critical component of many spatial biology analysis pipelines; a single foundation model that generalizes across cell morphologies and imaging methods would fill a crucial gap in modern biological workflows by expanding the scope of the data that can be processed. In this section, we demonstrate how the same CellSAM-generalist model (not fine-tuned to any particular dataset) can be used across biological imaging pipelines by highlighting two use cases: spatial transcriptomics and live-cell imaging (Fig. 3).

Spatial transcriptomics methods measure single-cell gene expression while retaining the spatial organization of the sample. These experiments (for example, MERFISH[83] and seqFISH[84]) fluorescently label individual mRNA transcripts; the number of spots for a gene inside a cell corresponds to that gene's expression level in that cell. These methods enable the investigation of spatial gene expression patterns from the subcellular to tissue scales but require accurate cell segmentation to

yield meaningful insights. Here, we use CellSAM-generalist in combination with Polaris[85], a deep-learning-enabled analysis pipeline for image-based spatial transcriptomics, to analyze gene expression at the single-cell level in MERFISH[86] and seqFISH[84] data (Fig. 3). With accurate segmentation, we can assign genes to specific cells (Fig. 3e). We note that CellSAM can perform segmentation on either images of nuclear and membrane stains or images derived from the spots themselves (for example, a maximum intensity projection of all spots). The ability of CellSAM to perform nuclear and whole-cell segmentation for challenging tissue images of dense cells with complex morphologies expands the scope of datasets to which Polaris can be applied.

The dynamics of cellular systems captured by live-cell imaging experiments elucidate various cellular processes such as cell division, morphological transitions and signal transduction[34]. The analysis of live-cell imaging data requires segmenting and tracking individual cells throughout whole movies. Here, we use CellSAM in combination with a cell tracking algorithm[87] (Fig. 3) in two settings. The first was a live-cell imaging experiment with HeLa cells transiently expressing an AMP kinase reporter[88] dosed with 20 mM 2-deoxy-D-glucose, a setup reflective of many experiments exploring cell signaling dynamics[13]. We imaged, segmented and tracked the cells over 60 frames or 120 minutes to quantify AMP kinase activity over time (Fig. 3b). The second setting was lineage tracking in budding yeast cells. We again used CellSAM and cell tracking to segment and track cells; we further used a division detection algorithm to count the cumulative number of divisions over time and trace individual cell lineages (Fig. 3c).

CellSAM can be used to segment three-dimensional images. We developed a workflow that generates three-dimensional segmentations by segmenting individual slices and aggregating them to three dimensions using the u-Segment3D[89] algorithm (Fig. 3d). We demonstrate on a thin slice of epidermal organoid[89] as well as EASI-FISH from the lateral hypothalamus[90].

The generality of CellSAM ensures consistent performance; in our experiments, we did not encounter catastrophic failure modes. This is in contrast to specialist models, where failures from inference on out-of-distribution data are common. CellSAM thus enables the analysis of data modalities for which specialist models do not exist (including any modality without widespread data). Furthermore, we simplify the analysis pipeline by eliminating the need for a large collection of potentially fragile specialist models. We note that use cases are representative of the modularity of bio-image analysis pipelines; most pipelines can be broken up into a few key steps. As CellSAM demonstrates, as the algorithms that perform these steps generalize, so does the entire pipeline.

## Discussion

Cell segmentation is a critical task for cellular imaging experiments. Although deep learning methods have made substantial progress in recent years, there remains a need for methods that can generalize across diverse images and further reduce the marginal cost of image labeling. In this work, we sought to meet these needs by developing CellSAM, a foundation model for cell segmentation. Transformer-based methods for cell segmentation have shown promising performance[91]. CellSAM builds upon these works by integrating the mask generation capabilities of SAM with transformer-based object detection to empower both scalable image labeling and automated inference. We trained CellSAM on a diverse dataset, and our benchmarking demonstrated that CellSAM achieves human-level performance on generalized cell segmentation. Compared to previous methods, CellSAM preserves its performance when trained on increasingly diverse data, which is essential for a foundational model. We found that CellSAM could be used on novel cell types in a zero-shot setting and that retraining with few labels could yield a strong boost in performance if needed. Moreover, we demonstrated that the ability of CellSAM to generalize can be extended to entire image analysis pipelines, as illustrated by use cases

in spatial transcriptomics and live-cell imaging. Given its utility in image labeling and high accuracy during inference, we think that CellSAM is a valuable contribution to the field, both as a tool for spatial biology and as a means to creating the data infrastructure required for cellular imaging's AI-powered future. To facilitate the former, we deployed a user interface for CellSAM at https://cellsam.deepcell.org/ that allows for both automated and manual prompting. We also created a Napari plugin so that CellSAM can be easily integrated into existing workflows.

We note that for generalist models to achieve wide adoption, equivalent or superior performance compared to specialist models is necessary. CellSAM is, to our knowledge, the first generalist model where this has been demonstrated across a diverse array of imaging data. This marks a critical threshold, as it enables users to adopt a unified model without sacrificing task-specific performance.

The work described here is relevant beyond aiding life scientists with cell segmentation. First, foundation models are immensely useful for natural language and vision tasks and hold similar promise for the life sciences, provided they are suitably adapted to this new domain. We can see several uses for CellSAM that might be within reach of future work. Much like what has occurred with natural images, we foresee that the integration of natural language labels in addition to cell-level labels might lead to vision–language models capable of generating human-like descriptors of cellular images with entity-level resolution[48]. More powerful generalization capabilities may enable the standardization of cellular image analysis pipelines across all the life sciences. If the accuracy is sufficient, microbiologists and tissue biologists could use the same collection of foundation models for interpreting their imaging data even for challenging experiments[92,93].

Although the work presented here highlights the potential that foundation models hold for cellular image analysis, much work remains to be done for this future to manifest. Extension of this methodology to three-dimensional imaging data is essential; recent work on memory-efficient attention kernels[94] will aid these efforts. Exploring how to enable foundation models to leverage the full information content of images (for example, multiple stains and temporal information for movies) is an essential avenue of future work. Although CellSAM generalizes well across many different cell types, performance guarantees can be made only for images similar to the training data. This was evidenced by our zero-shot evaluation on LIVECell. Although foundation models hold much promise, care must be taken with claims surrounding generalization. The most reliable way of obtaining generalization is in building the support of the training data distribution; the human-in-the-loop capabilities supported by SAM-like models can aid this effort. Generally, expanding the space of labeled data remains a priority; this includes images of perturbed cells and cells with more challenging morphologies (for example, neurons) where images contain many overlapping cells. Data generated by pooled optical screens[95] may synergize well with the data needs of foundation models. Compute-efficient fine-tuning strategies must be developed to enable flexible adaptation to new image domains. Lastly, prompt engineering is a critical area of future work, as it is essential to maximizing model performance. The work we present here can be thought of as prompt engineering, as we leverage CellFinder to produce bounding box prompts for SAM. As more challenging labeled datasets are incorporated, the nature of the 'best' prompts will likely evolve. Finding the best prompts for these new data is a task that will likely fall on both the computer vision and life science communities.

## Online content

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

## Methods

### Dataset construction

To train CellSAM, we combined 10 separate datasets spanning a variety of modalities: TissueNet[28], DeepBacs[64], BriFiSeg[65], Cellpose[25,26], Omnipose[66,67], YeastNet[68], YeaZ[69], the 2018 Kaggle DSB[70], a collection of H&E datasets[71–77] and an internally collected dataset of phase microscopy images across eight mammalian cell lines (Phase400). The LIVECell[63] dataset was held out for zero-shot/few-shot tests. Our collective dataset included images across multiple imaging modalities (brightfield, phase contrast, fluorescence and mass cytometry), imaging targets (histology sections, yeast, cell culture, bacteria and nuclei), length scales and morphologies. We did not do any preprocessing and left pixel intensities untouched. We treated nuclear and whole-cell channels as green and blue channels in an RGB image, respectively, and the red channel was always blank. We moved the green channel to blue for nuclear-only datasets (that is, BriFiSeg and DSB) to keep the blue channel always occupied.

If available, we used predetermined train/validation/test splits for each dataset; otherwise, we introduced 80%–10%–10% data splits. For datasets with multiple FOVs of the same object set, we required all FOVs to belong to the same split. We deferred all duplicated samples to the train split for published datasets with a preexisting data leak (detected by pixel-wise hashing). Our assembled dataset kept all images in their original size. We followed a widely used annotation scheme for labeling our masks, with zero representing the background and unique positive integers representing different objects. Although this format precludes accurate segmentation of overlapping objects, labels of this kind were not present in the dataset we compiled. We filtered out invalid cell labels if the label contained disjoint regions or if the label had only a 1-pixel height or width. The processed images with filtered annotations were used for training, validation and testing. We conducted some additional processing for LIVECell[63]. We converted annotations from the COCO format to the same labeling format that we used on the other datasets for consistency. We used Cellpose's[26] preprocessing function `livecell_ann_to_masks()` to remove overlapping regions. In addition, we noticed inconsistencies in ground truth labels as previously observed by the Cellpose team (see Fig. 1c in ref. [26]). We thus manually inspected the LIVECell test split to divide the annotation quality into three classes: good, medium and poor. We randomly selected images in the good split of the test set for the CellSAM few-shot learning task.

We modified our preparation pipeline for the NeurIPS challenge dataset. In addition to our standard preprocessing, we reduced bright spots by linearly rescaling the raw pixel intensities such that the 99.9 percentiles corresponded to 1.0. We then normalized each image with contrast-limited adaptive histogram equalization (CLAHE)[96] with a kernel size of 128 pixels. Our assembled NeurIPS dataset used a fixed image size of 512 × 512 pixels. Images shorter than 512 pixels on either axis were zero-padded up to 512. For images with more than 512 pixels on either axis, we tiled them to 512 × 512 pixels with a 25% overlap and filled the empty regions with zeros. Any cropped images without valid annotations were removed. All images were upsampled to 1,024 × 1,024 as input for CellSAM. The NeurIPS training dataset includes all train/validation/test splits from our standard datasets, plus the NeurIPS training and tuning datasets. We used NeurIPS open test set for validation and the hidden test set for the performance report.

Statistics on our full dataset are in Supplementary Table 1. To aid reproducibility, the dataset labels in Supplementary Table 1 correspond to the labels present in our compiled dataset available at https://cellsam.deepcell.org.

### CellSAM architecture

We adapted Anchor DETR[62] for the object detector for CellSAM (CellFinder). This choice was motivated by Anchor DETR being NMS[97] free. NMS suppresses bounding boxes with a high amount of overlap to remove duplicate detections. Although this works well for natural images, cellular images often have tightly clustered objects, and NMS-based methods such as the R-CNN family[80,81] can suffer from a low recall in this setting. We replaced the Anchor DETR's ResNet[98] backbone with the ViT[44] from the SAM model[52]; specifically, we used the base-sized ViT (ViT-B).

As the maximum number of cells per image is generally no more than 1,000, we increased the number of queries $q$ to 3,500, 3.5 times the maximum number of cells, based on Fig. 12 in DETR[99], which provided an estimate of the number of queries needed for a DETR method to detect all objects. We used one pattern $p$ for the Anchor generation as most objects in cellular detection are usually of similar scale.

**Training CellFinder**: We used a base learning rate of $10^{-4}$ for the Anchor DETR head and $10^{-5}$ for the SAM-ViT backbone. We used weight decay of $10^{-4}$ and clip norm of 0.1. We used AdamW[100] with a step-wise learning rate scheduler that drops the learning rate by a factor of 10 after the 1,960th epoch. We trained CellFinder for 2,800 epochs with a batch size of four across eight H100 GPUs.

**Fine-tuning CellSAM**: After we trained CellFinder with the SAM-ViT backbone, the SAM-ViT output features were no longer aligned with the rest of the model (that is, the prompt encoder and mask decoder). To close this distribution gap, we froze the SAM-ViT (such that it continues to function well with CellFinder) and trained the neck of the SAM model. We trained this neck using ground truth bounding boxes as inputs and segmentation masks of individual cells as targets. We used a learning rate of $10^{-4}$ and a weight decay of $10^{-4}$ for this training. We also used AdamW[100] for this training and did not clip the gradient.

We found this training recipe to be best with respect to computational cost and overfitting, based on small-scale ablation experiments.

### Inference

At inference, we followed the following workflow. First, the input was passed through the Anchor DETR fine-tuned ViT-B. This resulted in an embedding dimension of 768. This embedding was then passed as an input to two parts of CellSAM: (1) the trained Anchor DETR module (CellFinder) and (2) the fine-tuned neck, which is a two-dimensional convolutional network reducing the embedding dimensionality further to 256. The bounding box outputs of CellFinder were then sent into the prompt encoder, resulting in the prompt embedding. The prompt embeddings and neck embedding were then passed to the mask decoder, which outputs pixel-wise probabilities for the cell and another IoU-based confidence value for the prediction as a whole. This results in a tensor of shape $N \times W \times H$, where $N$ corresponds to the number of cells predicted. This tensor was processed with a sigmoid and a threshold operation, resulting in binarized images. Depending on the metric used, either we used this tensor directly with the $N$ scores (specifically for computation of the COCO AP @ 0.5 IoU) or we computed the argmax over the cell dimension $N$ to generate a tensor $W \times H$, where each pixel corresponded to a unique integer label for each cell.

**Thresholding**: Given CellSAM's model architecture, we had three different thresholds at inference time. First, we had a threshold on the bounding boxes generated by CellFinder, which we set to 0.4 across all datasets. We dynamically adjust this threshold by clustering the confidence values. We perform $k$-means clustering ($k = 2$) for the box confidences of all cells for a given image. We then compute the mean of both clusters ($T_\mu$)—that is, the separation between likely noncells and cells. We then adjust the final bounding box threshold as $T_{\text{box}} = \frac{2}{3} \times 0.4 + \frac{1}{3} \times T_\mu$. The resulting boxes were passed through the Mask Decoder. We had an overall mask score outputted by the IoU prediction head of the Mask Decoder, which we set to 0.5. Lastly, we thresholded the mask decoder output after applying the sigmoid function to each pixel, which we set at 0.5.

**CellSAM postprocessing**: We used the same postprocessing steps that are used by Cellpose[32]. This consisted of hole filling and island removal for each predicted cell.

**Inference time**: We compute the inference time as a function of the number of cells (Supplementary Fig. 1). To generate this plot, we randomly sampled two images from each dataset in our test set and filtered them to ensure a diverse range of cell counts. All benchmarks were performed on a single A6000 GPU. Unlike Cellpose, which requires only a single model pass, CellSAM performs one pass for each query (that is, each cell). As a result, the wall clock timescales (approximately) linearly with the number of cells. We also compare CPU and GPU inference times for both CellSAM and Cellpose on $n = 20$ randomly sampled images. Whereas, for GPU-based inference, both CellSAM and Cellpose take less than 1 second per image, Cellpose takes on average almost 8 seconds per image on CPU and CellSAM close to 12 seconds on CPU.

**Model implementation and training.** CellSAM was implemented in PyTorch[101]. For CellFinder, we modified the official Anchor DETR repo (https://github.com/megvii-research/AnchorDETR). For CellSAM, we modified the official Segment Anything repo (https://github.com/facebookresearch/segment-anything). We used PyTorch Lightning[102] to scale the training. Prototyping was done using NVIDIA's RTX 4090. We used machines with NVIDIA A6000s, A100s (40-GB and 80-GB versions) or H100s for the experiments in the paper. The overall training time for CellSAM-generalist was 6 days, 12 hours on 8× A100 GPUs for training CellFinder and an additional 1 day for fine-tuning the complete CellSAM model (note: fine-tuning was done on a 1× 4090 GPU). For the specialist models, these numbers can be linearly scaled by the fraction of the total data used (for example, for 1/9 of the time for 1/9 of the data). Below, we added a plot of the training curves for training CellFinder and CellSAM-generalist (Supplementary Fig. 2).

### Three-dimensional integration
We use u-segment3D[89] to generate three-dimensional cellular segmentations. We first segment each two-dimensional slice using the CellSAM-generalist model. We then remove small masks and fill holes using Cellpose's postprocessing. We then use the u-segment3D[89] indirect method to fuse the two-dimensional segmentations to three dimensions.

### Benchmarking
We benchmarked the performance of CellSAM models against Cellpose[25,26] trained on our compiled datasets.

**Cellpose model training.** We utilized the command provided by the Cellpose developers to train our custom Cellpose models. Using Cellpose library version 3.0.11, we trained both specialist and generalist models from scratch[26]. `python -m cellpose –train –train_size –use_gpu –dir{}–test_dir{}–img_filter_img–mask_filter_masks –pretrained_model None –chan 3 -chan2 2`

We kept all the hyperparameters untouched. We used the SGD optimizer with a weight decay of $10^{-5}$ and a batch size of 8. We trained each model for 500 epochs with a base learning rate of 0.2. We used the default learning rate scheduler. The learning rate increased linearly from 0 to 0.2 over the first 10 epochs and then decreased by a factor of 2 every 10 epochs after the 400th epoch. We trained each model on a single NVIDIA A6000 GPU. In total, we trained 10 specialist models and one generalist model. When evaluating the performance of the Cellpose model, we followed the implementation of the cyto3 model evaluating pipeline. We first estimated the cell diameters per image using a trained size model. Then, we predicted the segmentation masks.

**Metrics.** We used the Metrics package in the Cellpose library[25,26]. Predictions that match the ground truth labels (determined by a mask IoU ≥ 0.5) are true positives (TPs), predictions with no matching ground truth labels are false positives (FPs) and ground truth labels without a valid match are false negatives (FNs). For the human/human and CellSAM/human comparisons, we used a lower IoU threshold of 0.3

to increase the size of our statistic (reflecting higher inter-rater variability). The CellSAM/human comparisons were made against all three annotators. CellSAM's predictions were compared to each annotator, and each annotator was compared to every other annotator for the human/human comparison. We computed the recall, precision and F1 scores using the following formulas:

- Recall: recall $= \frac{TP}{TP+FN}$
- Precision: precision $= \frac{TP}{TP+FP}$
- F1: F1 $= \frac{2 \times precision \times recall}{precision+recall}$

We also used the COCO evaluation metrics[79] during CellFinder's development. The COCO metrics are a widely used benchmark for assessing the object-level quality of object detection and instance segmentation methods. These metrics report average precision—the area under the precision-recall curve for a given object class. In our case, we had only a single object class: cells. The average precision is computed for different IoU thresholds, ranging from 0.5 to 0.95, with a step size of 0.05. We report the mean average precision across all IoU thresholds, denoted as mAP, as well as the average precision at IoU = 0.5, denoted as AP50, to quantify CellFinder's performance. Because the object density is much higher in cellular images than in natural images, we modified the limit for the maximum number of detections from 100 to 10,000. We also fed the actual confidence score per binary prediction of the CellSAM model to the COCO evaluator. For the Cellpose models, we used a fixed confidence score of 1.0.

### Hyperparameters
For training, there are two sets of parameters. The first is for training CellFinder, which we trained for 2,800 epochs with a batch size of four (primarily constrained by GPU memory) across eight GPUs. We used the AdamW optimizer with a learning rate of $10^{-5}$ for the backbone and $10^{-4}$ for the remainder of the model. We set weight decay $10^{-4}$ and dropout to 0.1. After 1,960 epochs, we reduced the learning rate by a factor of 10. The full configuration file can be found in the GitHub repository.

The second set of parameters is fine-tuning CellSAM, which we trained for 50 epochs with a cosine learning rate schedule. This final fine-tuning primarily adjusts smaller components of CellSAM specifically, the 'neck' (a feedforward neural network between the ViT and the rest of the model), which is prone to overfitting and, thus, requires careful tuning.

### Reporting summary
Further information on research design is available in the Nature Portfolio Reporting Summary linked to this article.

### Data availability
All datasets with test/training/validation splits are publicly available at https://cellsam.deepcell.org.

### Code availability
The code for CellSAM is publicly available at https://github.com/vanvalenlab/cellsam.

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

## Acknowledgements

We thank L. Keren, N. Greenwald, S. Cooper, J. Funke, U. Manor, J. Horsman, M. Baym, P. Blainey, I. Cheeseman, M. Leonetti, N. Kondapaneni and E. Cole for valuable conversations and insightful feedback. We thank F. Zhou for helping with the three-dimensional segmentation. We also thank W. Graf, G. Miller, T. Dougherty, M. Schwartz and K. Yu, whose time in the Van Valen laboratory established the infrastructure and software tools that made this work possible. We thank N. Khalil, H. Carroll, A. Fong and the entire Brev.dev team for their support in establishing the computational infrastructure required for this work. We thank B. Strauch for help with figures and illustration. We also thank R. J. Xu and J. Moffitt for providing unpublished MERFISH data for the spatial transcriptomics workflow. We utilized images of the HeLa cell line in this research. Henrietta Lacks and the HeLa cell line established from her tumor cells without her knowledge or consent in 1951 have substantially contributed to scientific progress and advances in human health. We are grateful to Lacks, now deceased, and the Lacks family for their contributions to biomedical research. This work was supported by awards from the Shurl and Kay Curci Foundation (to D.V.V.), the Rita Allen Foundation (to D.V.V.), the Susan E. Riley Foundation (to D.V.V.), the Pew-Stewart Cancer Scholars program (to D.V.V.), the Gordon and Betty Moore Foundation (to D.V.V.), the Schmidt Academy for Software Engineering (to S.L.), the Michael J. Fox Foundation through the Aligning Science Across Parkinson's consortium (to D.V.V.), the Heritage Medical Research Institute (to D.V.V.), the National Institutes of Health (NIH) New Innovator program (DP2-GM149556) (to D.V.V.), the NIH HuBMAP consortium (OT2-OD033756) (to D.V.V.), the Howard Hughes Medical Institute Freeman Hrabowski Scholars program (to D.V.V.), the NIH (R01-MH123612A) (to P.P.), the NIH/Ohio State University (R01-DC014498) (to P.P.), the Chen Institute (to P.P.), the Emerald Foundation and Black in Cancer (to U.I.) and the Caltech Presidential Postdoctoral Fellowship Program (to U.I.).

## Author contributions

M.M. and U.I. contributed equally to this project and have the right to list themselves first in bibliographic documents. R.D. and Q.L. contributed equally to this project and have the right to list themselves second in bibliographic documents. M.M., U.I., Y.Y. and D.V.V. conceived the project. M.M., U.I., Q.L., Y.Y. and D.V.V. performed algorithm design for CellFinder and CellSAM. M.M. implemented the CellSAM architecture. M.M., U.I. and Q.L. implemented CellFinder. M.M., U.I., and Q.L. carried out the experiments and evaluations of the method. G.G. and P.P. provided input for developing CellFinder. Q.L. and U.I. performed model benchmarking. Q.L. and R.D. developed data pipelines. R.D. developed the computational infrastructure for model training. R.D., E. Pradhan, E. Pao, Q.L., C.Y. and E.L. performed data engineering. E.L., C.Y. and U.I. performed CellSAM integration with bioimaging workflows. R.D., A.P.-G., M.M. and R.B. developed the web portal and Napari plugin. M.M. performed three-dimensional CellSAM integration. A.A., M.A. and C.B. performed annotations on images for human/human comparison. R.B. and D.V.V. supervised the software engineering. D.V.V. supervised the project.

## Competing interests

D.V.V. is the scientific founder of Aizen Therapeutics and holds equity in the company. The other authors declare no competing interests.

## Additional information

**Extended data** is available for this paper at https://doi.org/10.1038/s41592-025-02879-w.

**Correspondence and requests for materials** should be addressed to David Van Valen.

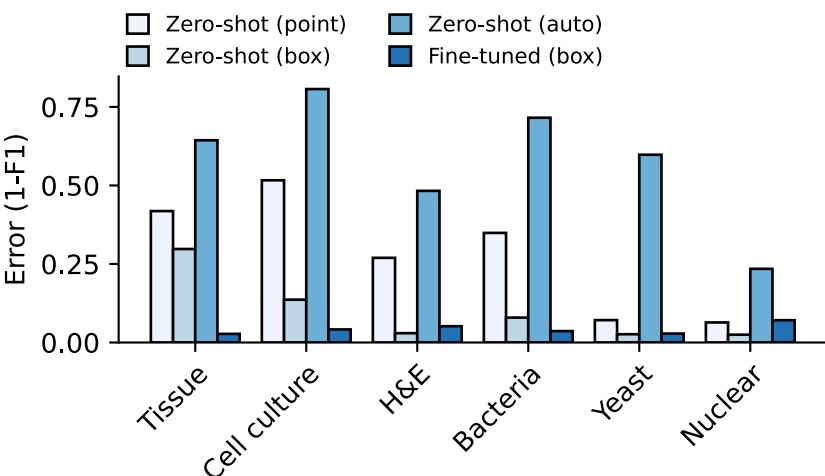

**Extended Data Fig. 1 | Preliminary prompting analysis.** Comparing point prompts (zero-shot point), bounding box prompts (zeroshot box), and SAM's auto prompting, a uniform grid of points, (zero-shot auto). We used the ground truth cellular masks to generate both the point and bounding box prompts (serving as a theoretical upper bound on performance). We used the F1 error (1-F1 score) for the metric of comparison. Additionally, we used SAM with no additional training to assess zero-shot performance and compared that to a fine-tuned SAM. Here, we see the best performance was achieved using bounding box prompts with a fine-tuned model. The bounding boxes we can generate using CellFinder in the full model. The categories contained the following number of samples (n): Tissue = 330, Cell Culture = 144, H&E = 51, Bacteria = 260, Yeast = 32, and Nuclear = 56.

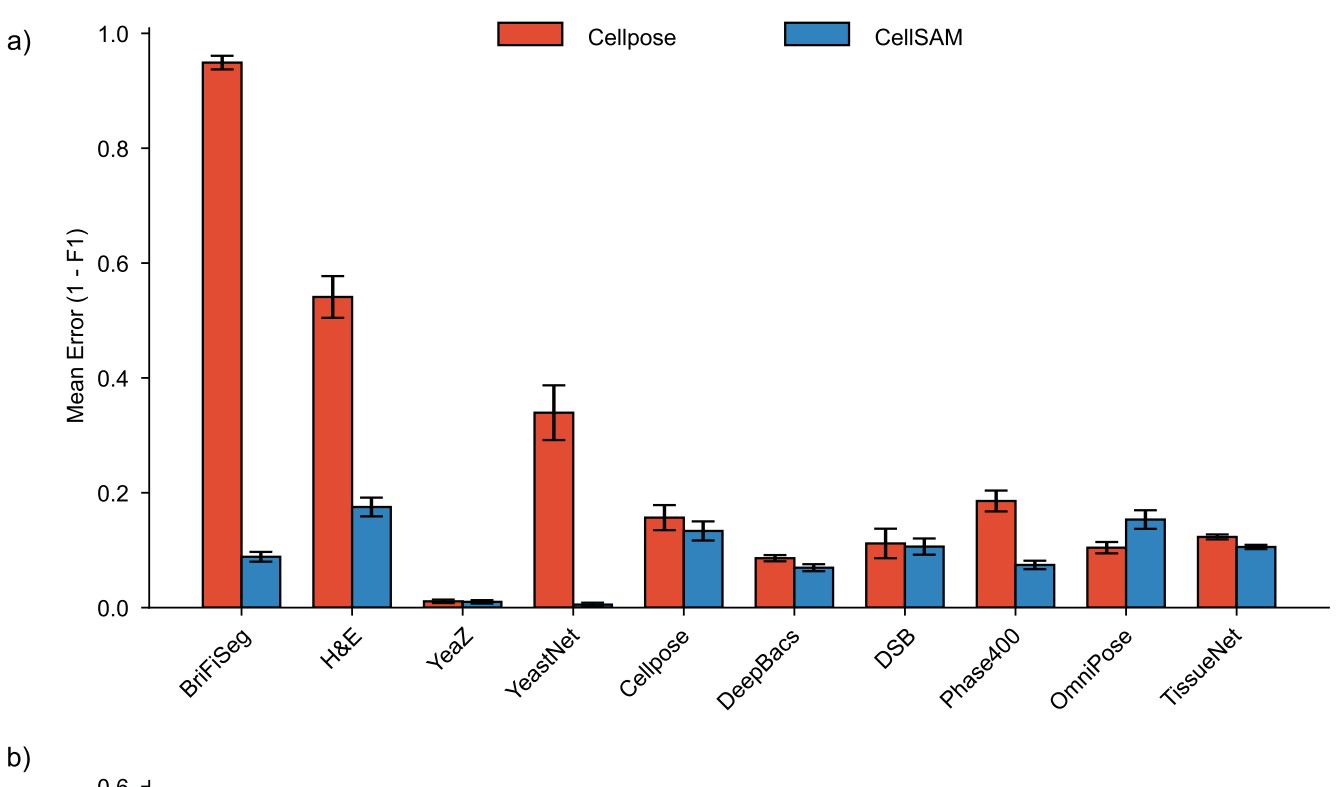

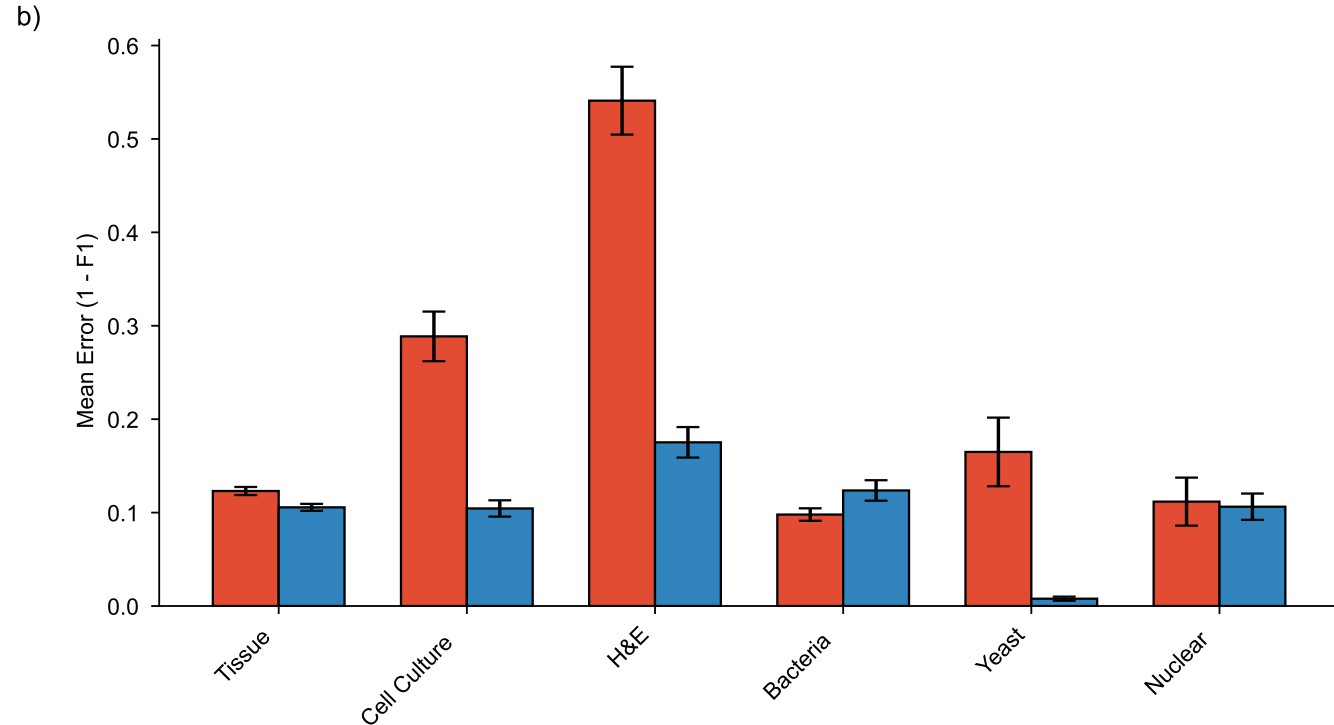

**Extended Data Fig. 2 | CellSAM-generalist Comapred to Cellpose generalist model (cyto3).** Here, we compare CellSAM generalist performance to Cellpose's cyto3 model. a) examines the per dataset performance of these two models. The datasets contained the following number of samples: BriFiSeg = 22, H&E = 51, YeaZ = 17, YeastNet = 15, Cellpose = 68, DeepBacs = 92, DSB = 56, Phase400 = 54, OmniPose = 168, and TissueNet = 330. b) examples the performance across the various data types. The categories contained the following number of samples: Tissue = 330, Cell Culture = 144, H&E = 51, Bacteria = 260, Yeast = 32, and Nuclear = 56.

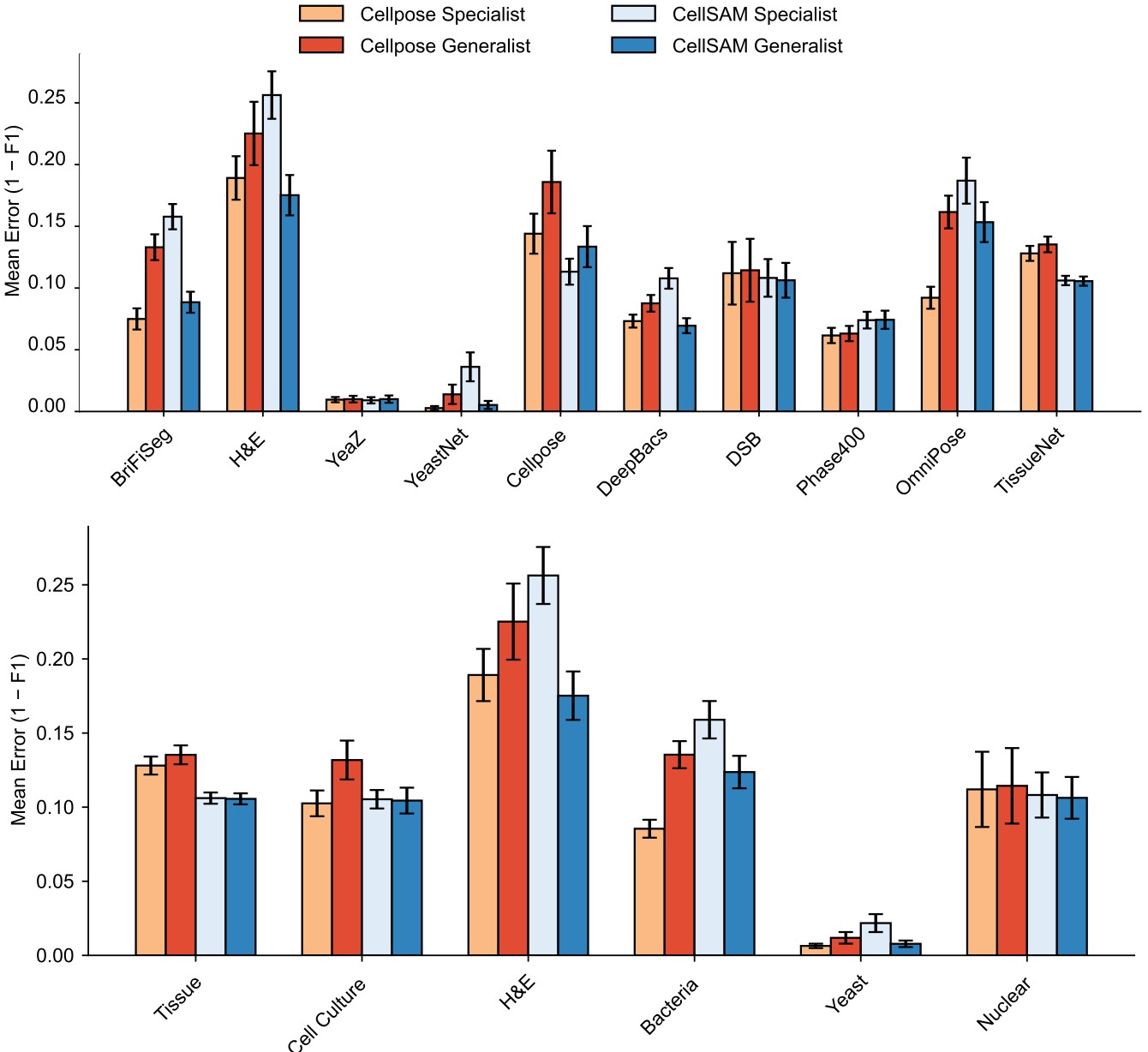

**Extended Data Fig. 3 | Generalist vs Specialist.** We compare the performance of CellSAM when trained on individual datasets (specialist) to when CellSAM is trained on all of the data (generalist). These results are compared to the performance of Cellpose when trained on individual datasets (specialist) and when Cellpose is trained on all the data (generalist). We see that CellSAM-generalist performs better across data types and datasets. Cellpose consistently performs better when trained on a specific dataset, but does not perform well when trained across multiple datasets.

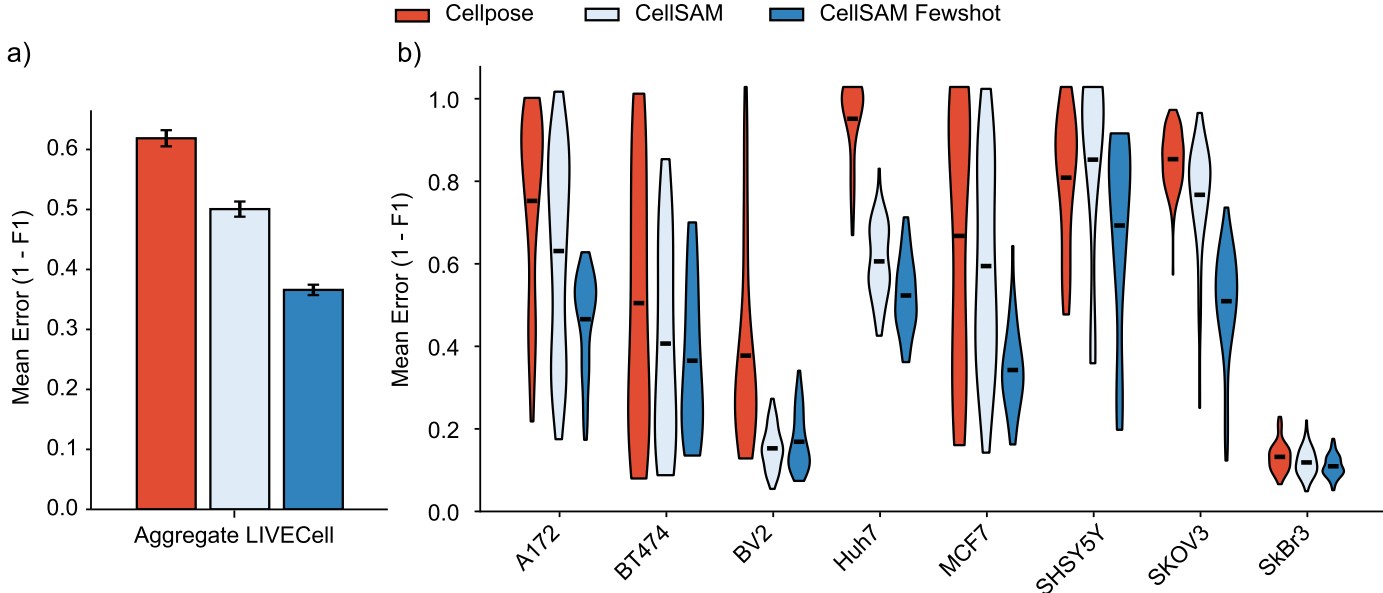

**Extended Data Fig. 4 | a**, Zero-shot performance of CellSAM-generalist and Cellpose-generalist on the LIVECell dataset. Here, we show significant improvement of CellSAM over Cellpose on an unseen dataset (from 0.13 to 0.40 in F1). **b**, CellSAM generalist performance stratified by cell line. We analyzed both zero-shot and few-shot (10 samples per cell line) performance. We saw that few-shot improves CellSAM-generalist on LIVECell for most cell lines.

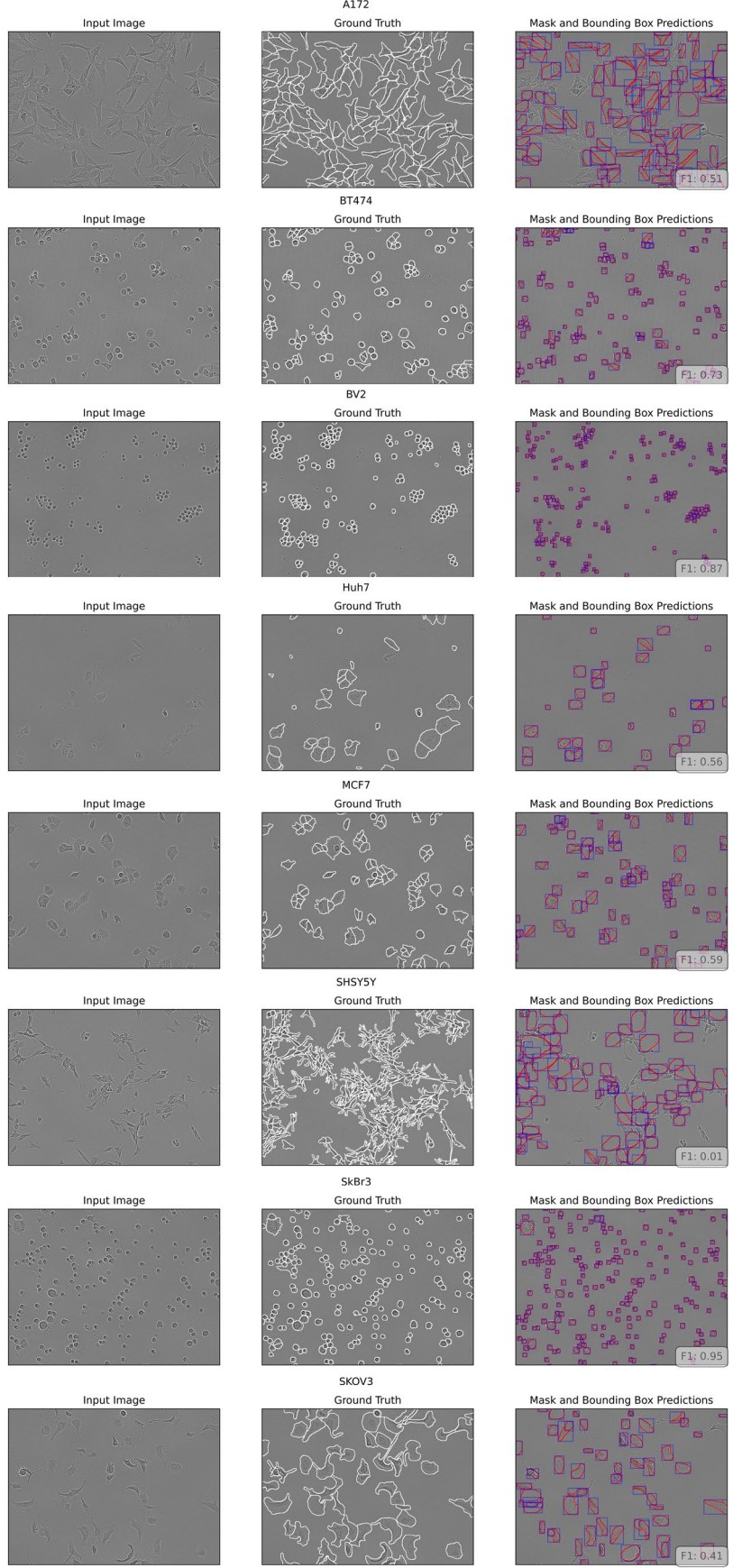

**Extended Data Fig. 5 | Representative qualitative results of CellSAM-generalist 10-shot performance on the LIVECell dataset, one panel for each cell line.

| | |
|---|---|

# Reporting Summary

Please do not complete any field with "not applicable" or n/a. Refer to the help text for what text to use if an item is not relevant to your study.
For final submission: please carefully check your responses for accuracy; you will not be able to make changes later.

## Statistics

For all statistical analyses, confirm that the following items are present in the figure legend, table legend, main text, or Methods section.

| n/a | Confirmed | |
|---|---|---|
| ☐ | ☐ | The exact sample size (*n*) for each experimental group/condition, given as a discrete number and unit of measurement |
| ☐ | ☐ | A statement on whether measurements were taken from distinct samples or whether the same sample was measured repeatedly |
| ☐ | ☒ | The statistical test(s) used AND whether they are one- or two-sided<br>*Only common tests should be described solely by name; describe more complex techniques in the Methods section.* |
| ☐ | ☐ | A description of all covariates tested |
| ☐ | ☐ | A description of any assumptions or corrections, such as tests of normality and adjustment for multiple comparisons |
| ☐ | ☒ | A full description of the statistical parameters including central tendency (e.g. means) or other basic estimates (e.g. regression coefficient) AND variation (e.g. standard deviation) or associated estimates of uncertainty (e.g. confidence intervals) |
| ☐ | ☐ | For null hypothesis testing, the test statistic (e.g. *F*, *t*, *r*) with confidence intervals, effect sizes, degrees of freedom and *P* value noted<br>*Give P values as exact values whenever suitable.* |
| ☐ | ☐ | For Bayesian analysis, information on the choice of priors and Markov chain Monte Carlo settings |
| ☐ | ☐ | For hierarchical and complex designs, identification of the appropriate level for tests and full reporting of outcomes |
| ☐ | ☐ | Estimates of effect sizes (e.g. Cohen's *d*, Pearson's *r*), indicating how they were calculated |

*Our web collection on statistics for biologists contains articles on many of the points above.*

## Software and code

Policy information about availability of computer code

| Data collection | |
|---|---|
| Data analysis | All data and code is available at github.com/vanvalenlab and cellsam.deepcell.org |

For manuscripts utilizing custom algorithms or software that are central to the research but not yet described in published literature, software must be made available to editors and reviewers. We strongly encourage code deposition in a community repository (e.g. GitHub). See the Nature Portfolio guidelines for submitting code & software for further information.

## Data

Policy information about availability of data

All manuscripts must include a data availability statement. This statement should provide the following information, where applicable:
- Accession codes, unique identifiers, or web links for publicly available datasets
- A description of any restrictions on data availability
- For clinical datasets or third party data, please ensure that the statement adheres to our policy

All data and code is available at github.com/vanvalenlab and cellsam.deepcell.org

## Research involving human participants, their data, or biological material

Policy information about studies with human participants or human data. See also policy information about sex, gender (identity/presentation), and sexual orientation and race, ethnicity and racism.

| | |
|---|---|
| Reporting on sex and gender | |
| Reporting on race, ethnicity, or other socially relevant groupings | |
| Population characteristics | |
| Recruitment | |
| Ethics oversight | |

Note that full information on the approval of the study protocol must also be provided in the manuscript.

# Field-specific reporting

Please select the one below that is the best fit for your research. If you are not sure, read the appropriate sections before making your selection.

☐ Life sciences  ☐ Behavioural & social sciences  ☐ Ecological, evolutionary & environmental sciences

For a reference copy of the document with all sections, see nature.com/documents/nr-reporting-summary-flat.pdf

# Life sciences study design

All studies must disclose on these points even when the disclosure is negative.

| | |
|---|---|
| Sample size | |
| Data exclusions | |
| Replication | |
| Randomization | |
| Blinding | |

# Behavioural & social sciences study design

All studies must disclose on these points even when the disclosure is negative.

| | |
|---|---|
| Study description | |
| Research sample | |
| Sampling strategy | |
| Data collection | |
| Timing | |
| Data exclusions | |
| Non-participation | |
| Randomization | |

# Ecological, evolutionary & environmental sciences study design

All studies must disclose on these points even when the disclosure is negative.

| | |
|---|---|
| Study description | |
| Research sample | |
| Sampling strategy | |
| Data collection | |
| Timing and spatial scale | |
| Data exclusions | |
| Reproducibility | |
| Randomization | |
| Blinding | |

Did the study involve field work?  ☐ Yes  ☐ No

## Field work, collection and transport

| | |
|---|---|
| Field conditions | |
| Location | |
| Access & import/export | |
| Disturbance | |

# Reporting for specific materials, systems and methods

We require information from authors about some types of materials, experimental systems and methods used in many studies. Here, indicate whether each material, system or method listed is relevant to your study. If you are not sure if a list item applies to your research, read the appropriate section before selecting a response.

## Materials & experimental systems

| n/a | Involved in the study |
|---|---|
| ☐ | ☐ Antibodies |
| ☐ | ☐ Eukaryotic cell lines |
| ☐ | ☐ Palaeontology and archaeology |
| ☐ | ☐ Animals and other organisms |
| ☐ | ☐ Clinical data |
| ☐ | ☐ Dual use research of concern |
| ☐ | ☐ Plants |

## Methods

| n/a | Involved in the study |
|---|---|
| ☐ | ☐ ChIP-seq |
| ☐ | ☐ Flow cytometry |
| ☐ | ☐ MRI-based neuroimaging |

## Antibodies

| | |
|---|---|
| Antibodies used | |
| Validation | |

# Eukaryotic cell lines

Policy information about cell lines and Sex and Gender in Research

| | |
|---|---|
| Cell line source(s) | |
| Authentication | |
| Mycoplasma contamination | |
| Commonly misidentified lines (See ICLAC register) | |

# Palaeontology and Archaeology

| | |
|---|---|
| Specimen provenance | |
| Specimen deposition | |
| Dating methods | |

☐ Tick this box to confirm that the raw and calibrated dates are available in the paper or in Supplementary Information.

| | |
|---|---|
| Ethics oversight | |

Note that full information on the approval of the study protocol must also be provided in the manuscript.

# Animals and other research organisms

Policy information about studies involving animals; ARRIVE guidelines recommended for reporting animal research, and Sex and Gender in Research

| | |
|---|---|
| Laboratory animals | |
| Wild animals | |
| Reporting on sex | |
| Field-collected samples | |
| Ethics oversight | |

Note that full information on the approval of the study protocol must also be provided in the manuscript.

# Clinical data

Policy information about clinical studies
All manuscripts should comply with the ICMJE guidelines for publication of clinical research and a completed CONSORT checklist must be included with all submissions.

| | |
|---|---|
| Clinical trial registration | |
| Study protocol | |
| Data collection | |
| Outcomes | |

# Dual use research of concern

Policy information about dual use research of concern

## Hazards

Could the accidental, deliberate or reckless misuse of agents or technologies generated in the work, or the application of information presented in the manuscript, pose a threat to:

| No | Yes | |
|----|-----|--|
| ☐ | ☐ | Public health |
| ☐ | ☐ | National security |
| ☐ | ☐ | Crops and/or livestock |
| ☐ | ☐ | Ecosystems |
| ☐ | ☐ | Any other significant area |

## Experiments of concern

Does the work involve any of these experiments of concern:

| No | Yes | |
|----|-----|--|
| ☐ | ☐ | Demonstrate how to render a vaccine ineffective |
| ☐ | ☐ | Confer resistance to therapeutically useful antibiotics or antiviral agents |
| ☐ | ☐ | Enhance the virulence of a pathogen or render a nonpathogen virulent |
| ☐ | ☐ | Increase transmissibility of a pathogen |
| ☐ | ☐ | Alter the host range of a pathogen |
| ☐ | ☐ | Enable evasion of diagnostic/detection modalities |
| ☐ | ☐ | Enable the weaponization of a biological agent or toxin |
| ☐ | ☐ | Any other potentially harmful combination of experiments and agents |

# Plants

Seed stocks

Novel plant genotypes

Authentication

# ChIP-seq

## Data deposition

☐ Confirm that both raw and final processed data have been deposited in a public database such as GEO.

☐ Confirm that you have deposited or provided access to graph files (e.g. BED files) for the called peaks.

Data access links
*May remain private before publication.*

Files in database submission

Genome browser session
(e.g. UCSC)

## Methodology

Replicates

Sequencing depth

Antibodies

Peak calling parameters

Data quality

| Software | |
|---|---|

# Flow Cytometry

## Plots

Confirm that:

☐ The axis labels state the marker and fluorochrome used (e.g. CD4-FITC).

☐ The axis scales are clearly visible. Include numbers along axes only for bottom left plot of group (a 'group' is an analysis of identical markers).

☐ All plots are contour plots with outliers or pseudocolor plots.

☐ A numerical value for number of cells or percentage (with statistics) is provided.

## Methodology

| Sample preparation | |
|---|---|
| Instrument | |
| Software | |
| Cell population abundance | |
| Gating strategy | |

☐ Tick this box to confirm that a figure exemplifying the gating strategy is provided in the Supplementary Information.

# Magnetic resonance imaging

## Experimental design

| Design type | |
|---|---|
| Design specifications | |
| Behavioral performance measures | |

| Imaging type(s) | |
|---|---|
| Field strength | |
| Sequence & imaging parameters | |
| Area of acquisition | |

Diffusion MRI    ☐ Used    ☐ Not used

## Preprocessing

| Preprocessing software | |
|---|---|
| Normalization | |
| Normalization template | |
| Noise and artifact removal | |
| Volume censoring | |

## Statistical modeling & inference

| Model type and settings | |
|---|---|
| Effect(s) tested | |

Specify type of analysis:  ☐ Whole brain  ☐ ROI-based  ☐ Both

Statistic type for inference

(See Eklund et al. 2016)

Correction

## Models & analysis

| n/a | Involved in the study |
| --- | --- |
| ☐ | ☐ Functional and/or effective connectivity |
| ☐ | ☐ Graph analysis |
| ☐ | ☐ Multivariate modeling or predictive analysis |

Functional and/or effective connectivity

Graph analysis

Multivariate modeling and predictive analysis

