## [Peer Review File · Nature Methods]

CellSAM: A Foundation Model for Cell Segmentation

Corresponding Author: Dr David Van Valen

Version 0:

Decision Letter:

23rd Apr 2024

Dear Dave,

Your Article entitled "A Foundation Model for Cell Segmentation" has now been seen by three reviewers, whose comments are attached. While they find your work of some potential interest, they have raised concerns which in our view are sufficiently important that they preclude publication of the work in Nature Methods.

We will consider looking at a revised manuscript only if further experimental data allow you to address all the major criticisms of the reviewers (unless, of course, something similar has by then been accepted at Nature Methods or appeared elsewhere). This includes submission or publication of a portion of this work somewhere else.

The required new experiments and data include, but are not limited to, updated benchmarking (ensuring fair comparisons to Cellpose and other approaches), further demonstrations on challenging datasets, and addition of requested data and clarifications. In addition, we would either need your team to make a stronger case that CellSAM truly is a foundation model, or for these claims to be removed.

If you are interested in revising this manuscript for submission to Nature Methods in the future, please contact me to discuss your appeal before making any revisions. Otherwise, we hope that you find the reviewers' comments helpful when preparing your paper for submission elsewhere.

Sincerely,
Rita

Rita Strack, Ph.D.
Senior Editor
Nature Methods

Although we cannot publish your paper, it may be appropriate for another journal in the Nature Portfolio. If you wish to explore the journals and transfer your manuscript please use our manuscript transfer portal. You will not have to re-supply manuscript metadata and files, unless you wish to make modifications. For more information, please see our [manuscript transfer FAQ](http://www.nature.com/authors/author_resources/transfer_manuscripts.html?WT.mc_id=EMI_NPG_1511_AUTHORTRANSF&WT.ec_id=AUTHOR) page.

Reviewers' Comments:

Reviewer #1:

Remarks to the Author:

A. Summary of the key results

In their paper, "A Foundation Model for Cell Segmentation", the authors present CellSAM, a method to perform instance segmentation of cells in 2D image data across a large variety of datasets and image modalities. CellSAM is based on prompt engineering by means of combining the original Segment Anything Model (SAM, Kirillov et al., 2023) with a bounding-box object detector. The object detector is an Anchor DETR (Wang, 2022) with SAM's encoder's vision transformer (ViT) as backbone.

The authors curated a diverse dataset to train CellSAM, made of nine existing public datasets (TissueNet, DeepBacs, BriFiSeg, Cellpose, Omnipose, YeastNet, YeaZ, the 2018 Kaggle Data Science Bowl dataset, and a collection of H&E datasets) along with their own dataset of phase microscopy images (Phase400) for cell and nuclei segmentation. In-domain evaluation was performed in held-out partitions of those datasets, while zero-shot and few-shot performance was evaluated in the public LIVECell dataset.

Specialist models were trained for six defined categories of images within the final dataset (Tissue, Cell Culture, H&E, Bacteria, Yeast and Nuclear), while a generalist model was trained on the full dataset. Their in-domain results are equivalent or improve over those of their own trained version of Cellpose on the same datasets. Remarkably, both specialist and generalist methods yield essentially equivalent results.

Finally, as use cases, the authors show two applications of CellSAM: one as an enabler of cell tracking and one as a key element in a transcriptomics analysis.

B. Originality and significance

As mentioned above, CellSAM is a combination of two existing state-of-the-art models: Anchor DETR (Wang, 2022) and SAM (Kirillov et al., 2023). Therefore, the novelty of this work lies primarily in using the ViT encoder of SAM as backbone network for the Anchor DETR and its combination to learn the instance segmentation task on a diverse dataset of cells and nuclei using automatic bounding boxes as prompts.

While I find this tool very useful for the bioimage analysis community, strictly speaking, I do not think it can be considered “a foundation model for cell segmentation” as stated by the authors. In the seminal paper about foundation models (“On the Opportunities and Risks of Foundation Models”, Bommasani et al, 2021), its authors clearly state that “A foundation model is any model that is trained on broad data (generally using self-supervision at scale) that can be adapted (e.g., fine-tuned) to a wide range of downstream tasks”. This definition diverges from the CellSAM approach, which was trained on broad data (although only cell data) but can only perform a single task (instance segmentation) with a single type of prompts (bounding boxes). In my opinion, the type of contribution that CellSAM represents was already defined in the original SAM paper: “An important distinction in our work is that a model trained for promptable segmentation can perform a new, different task at inference time by acting as a component in a larger system, e.g., to perform instance segmentation, a promptable segmentation model is combined with an existing object detector.”. Following this rationale, CellSAM should be recognized as the composite system that executes instance segmentation via the fusion of a segmentation foundation model (SAM) and prompt engineering in the form of a cell-specialized object detector (CellFinder). Hence, labeling CellSAM as a foundation model could lead to misconceptions among potential users, as it may imply versatility across diverse tasks, which is not currently feasible.

I have a couple of observations regarding the experiments performed to compare zero-shot performance using points and bounding boxes with either the default SAM or (fine-tuned) CellSAM (Figures S1 and S3). First, the fact that bounding box prompting performs better than point prompting had previously been demonstrated on biomedical image data (see Huang et al, 2024, and Archit et al, 2023, references [49] and [56] in the manuscript). Second, although the fine-tuned approach commonly outperforms the other approaches in most cases, it does not improve on the zero-shot bounding box performance on three out of the six categories in Fig. S3. On one hand, the improved performance of the fine-tuned model compared to the original model is an anticipated outcome, given the model's training on more specific and relevant data. This aligns with findings reported in the Cellpose 2.0 paper (Pachitariu & Stringer, 2022), where it was demonstrated that pretrained models only need a reduced subset of training data to achieve the same performance improvement as with the full training dataset. On the other hand, the strong performance of the non-fine-tuned zero-shot approach raises questions about the need of fine-tuning itself and whether equivalent results could be achieved through improved prompting strategies alone.

Regarding section 2.4 (“CellSAM unifies biological image analysis workflows”), I believe the text needs some clarifications. First, is the model used in both workflows the same (a generalist CellSAM) or different (a specialist CellSAM)? And moreover, what makes CellSAM unique to unify workflows as opposed to alternative generalist methods? In other words, it is unclear what the added value of CellSAM is compared to an approach as Cellpose in that scenario.

C. Data & methodology: validity of approach, quality of data, quality of presentation

The authors have conducted an extensive validation of their tool trained on a total of nine publicly available datasets plus their own collection of phase microscopy images. An extra public dataset (LIVECell) was used to evaluate the zero-shot and few-shot contexts. Their evaluation involved (1) benchmarking different zero-shot strategies against the original SAM model, that serves as reference for comparison with their own fine-tuned models, and (2) benchmarking specialists and generalist models of CellSAM against equivalent Cellpose models trained on the same datasets.

For the evaluation of segmentation results, the authors use a suite of metrics from their own DeepCell library. Namely, prediction, recall and F1 score (considering true positives any match with more than 0.6 of IoU with the ground truth labels). Additionally, the AP50 metric from COCO was also used to evaluate their detection method. In the figures, they define a general “error” metric as 1 minus a specific metric, setting an admissible real-world error value for error = 1-F1 under 0.2. While informative, the selection rationale for this value could benefit from further elaboration to provide clarity to readers.

Instance segmentation results are presented both quantitatively (based on their selected segmentation metrics) and qualitatively. For their 10-shot experiments in the LIVECell datasets, some quantitative and qualitative results are shown per cell line (Fig. S5). In that case, the visualization of results could be improved by showing bounding boxes and using a sparser color map for the segmentation masks.

An aspect that warrants attention is the absence of an evaluation of the usability of the proposed approach. Notably, there is no mention of inference times for either CellSAM or Cellpose. Particularly in the few-shot scenario, understanding the tool and time investment required for annotating the limited samples would provide valuable insights into the practicality of the approach.

D. Appropriate use of statistics and treatment of uncertainties

The authors made a correct use of statistical tests to check the significance of the differences between the performance of CellSAM and that of a human annotator and those between two human annotators in different datasets. As mentioned above, my only suggestion in this aspect is the inclusion of an explanation on the selection of 0.2 as the value to consider an error performance acceptable in a real-world scenario.

E. Conclusions: robustness, validity, reliability

The authors have introduced CellSAM, a method that builds upon the original SAM framework to facilitate automatic instance segmentation of cells in 2D images. Their comparative results against their own trained version of Cellpose demonstrate competitive performance across both in-domain and zero-shot scenarios. Particularly noteworthy is the observation that the generalist CellSAM, trained on all datasets, performs equivalently to the specialist CellSAM models trained on specific datasets, a distinction not observed in Cellpose.

As previously discussed, while I maintain reservations about labeling CellSAM as a foundation model based on its adherence to the original definition by Bommasani et al. and its singular task and prompt type, I believe CellSAM constitutes a valuable addition to the segmentation toolkit for the bioimage community. Furthermore, enhancing its usability through the implementation of a more interactive interface could position it as a compelling alternative to Cellpose.

It's important to acknowledge that the evaluation of CellSAM against the state of the art is currently limited to comparisons with their own trained versions of Cellpose. This approach may raise some controversy, as highlighted in recent discussions within the field (refer to Stringer et al., bioRxiv, 2024, <https://doi.org/10.1101/2024.04.06.587952>).

F. Suggested improvements: experiments, data for possible revision

From a methodological point of view, I believe the manuscript, and the project in general, would benefit from the following improvements:

- 1) Further documentation of the training and evaluation processes: It would be beneficial to provide a more detailed account of the training process of both CellSAM and Cellpose models. In particular, I would:
 - a. Follow the specifications recommended by Dodge et al. ("Show your work: Improved reporting of experimental results", arXiv preprint arXiv:1909.03004, 2019) to describe all the training configurations and hyperparameters tested on each case.
 - b. Display of training and validation plots. To enhance transparency and understanding of the training process, consider including training and validation plots for key metrics such as loss and any proxy metric (Dice, Jaccard, etc.). These plots can offer readers insights into the convergence behavior of models, helping them gauge when and how the models reached optimal performance.
 - c. Provide training times for the final models to offer readers insights into the computational implications of different settings and experiments.
 - d. Provide inference/testing times per full-size image. Given equivalent or similar performance, some potential users may decide on the model to use based on these numbers depending on the experimental scenario. For instance, this could be crucial in large- or small-scale setups.
 - e. Offer more details about the few-shot data preparation and retraining settings. Namely, which tool is used for annotating new samples? Are those annotations dense or sparse? Which are the annotation times for both CellSAM and Cellpose (if added to the comparison)? What are the hyperparameters used for fine-tuning with the new few annotations?
 - f. Provide a better description of the assembled dataset by including the number of images and cells per subset. This would help better understand the training process and the possible impact of imbalances with respect to cell types and imaging modalities.
- 2) Extend the comparison with other state-of-the-art (SOTA) methods for the in-domain setting. Given the diverse dataset employed to build CellSAM generalist and specialist models, why not use the actual SOTA tables from the original datasets to compare their performance with other existing approaches when possible? This would help readers to better place CellSAM's performance in context with specialized methods that have been optimized for each dataset.
- 3) Expand the comparison with Cellpose using the NeurIPS challenge dataset. To avoid any controversy while comparing CellSAM's performance against Cellpose, it would be both beneficial and very interesting for the target audience to use a dataset for which Cellpose authors have optimized their own method. Moreover, using the NeurIPS challenge dataset (<https://neurips22-cellseg.grand-challenge.org/>) has the added value of importing an extra set of SOTA methods that can be included automatically for comparison. I truly believe the potential readers of the manuscript would appreciate putting the performance of CellSAM in that context, which has very recently become a hot topic in the field.
- 4) Improve CellSAM's usability with interactive tools. Enhance CellSAM's usability by integrating it with established bioimage analysis, visualization, and processing tools such as Fiji or Napari. Additionally, consider developing an interactive tool specifically designed for proofreading or few-shot scenarios to streamline user workflows and improve user experience.
- 5) Expand the repository to provide training code. Expand the existing repository to include training scripts for the sake of completeness and reproducibility, enabling users to replicate experiments and further contribute to the project's development.

Minor comments:

- * All over the text, please, homogenize the naming of key terms such as “foundation” (sometimes referred to as “foundational”), “generalist” (sometimes referred to as “general”) and “specialist” (sometimes referred to as “specific”).
- * On page 2, the text reads “Cell segmentation is a variant of the instance segmentation problem”, which seems confusing. My understanding is that cell segmentation is just one of the multiple applications of instance segmentation, not a “variant” of it.
- * In Fig. 1, I would clearly specify in the drawing that the ViT is part of the original SAM, as it is done with the decoder.
- * On page 6, at the end of the page there are two typos:
 - ** “we observed that CellSAMpreserved its” => “we observed that CellSAM preserved its”.
 - ** “difference in F1 scores was \geq 0.05,” => “difference in F1 scores was $<$ 0.05.”.
- * On page 7, there is a missing period after “CellSAM enables fast and accurate labeling”.
- * In Fig. 3, specify what CellSAM model was used in both settings (generalist or specialist). There is also what seems to be an unfinished sentence in the caption: “(b) Application of CellSAM to various biological imaging.”
- * Revise the text of the references, some of them are missing their respective journals of publication (See for instance [34], [43], [44]...).
- * In Appendix A, the size of the dataset images is said to be 512x512 pixels, but SAM’s encoder has an input of 1024x1024. I assume you’re upsampling your images before going through the ViT but, please, confirm that in the text.
- * On page 19, you say that you “manually inspected the LIVECell test split to divide the annotation quality into three classes - good, medium, and poor. We randomly selected images in the good split of the validation set for the CellSAM few-shot learning task.” This is a bit confusing. Did you divide the validation set as well into three classes or did you use the test set as a validation set?
- * In Fig. S1, there are two typos: “We demonstrated CellSAM-specificand CellSAM-generalsuperior performance” => “We demonstrated CellSAM-specific and CellSAM-general superior performance”. Notices as well “specific” and “general” should be replaced by “specialized” and “generalist”.
- * In Fig. S1 as well, considered defining the “ZS point”, “ZS Box” and “FT Box” in the caption and using more distinct colors for the bar plots.
- * In Fig. S3, please specify on which data is the fine-tuning performed. I assume you’re referring to the CellSAM generalist model, so it was trained on all the training partitions of the original dataset. Moreover, if I understand correctly the plot, the results using zero-shot and bounding boxes are better than using fine tuning and the ground truth boxes for the nuclear category. How is that possible?
- * In Fig. S4, there are some typos (“CS GD Bbox”) and many undefined acronyms. Also, the colors are not assigned to the same experiments in (a) and (b).
- * The conclusion extracted from Fig. S4a, (“CellSAM can considerably improve its performance with only ten additional fields of views”) seems a bit an overstatement. The improvement seems very marginal in that plot.

G. References

The manuscript appropriately acknowledges previous work through accurate and well-placed citations throughout the text.

H. Clarity and context: lucidity of abstract/summary, appropriateness of abstract, introduction and conclusions

The text of the reviewed paper exhibits a high level of clarity, ensuring accessibility for readers within the scientific community. The language employed is precise, technical, and apt for the subject matter, facilitating comprehension of the research methodology and findings. Additionally, the paper maintains a logical and well-structured flow, guiding readers through the study with ease. To further enhance its quality, attention to defining technical terms, providing contextual information, and offering detailed explanations of the experimental setup would contribute to a more informative and comprehensive paper.

However, it's important to address concerns regarding certain aspects within the manuscript. Specifically, characterizing CellSAM as a foundation model may not align with its training on a specific dataset for a single task, as outlined in Bommasani et al. (2021). Similarly, the assertion that CellSAM unifies biological image analysis workflows lacks substantial support within the manuscript. These points should be carefully reviewed and potentially clarified to ensure the accurate representation of the method's capabilities and contributions.

Reviewer #2:

Remarks to the Author:

The presented paper proposes a generalist approach to cell segmentation, which builds on top of the “Segment Anything (SAM)” approach proposed by Meta AI. A DETR-based detection approach is used to obtain bounding box annotations automatically in a preceding step to facilitate the application of the SAM segmentation approach and offer a fully-automated processing pipeline. The authors demonstrate that the combination of both approaches, forming the proposed CellSAM pipeline, leads to precise segmentation results on various datasets. Different experiments are conducted to examine the generalizability and fine-tuning ability in comparison to a state-of-the-art segmentation approach, Cellpose. The reported results show that CellSAM outperforms the state-of-the-art and reaches human-level accuracy. The paper is well-structured and concise; nevertheless, I have some remarks, which are listed in the following.

Major Points:

- As CellFinder is meant to prevent the necessity for human input, it might be interesting from a practical point of view to emphasize whether there is a trade-off between accuracy and automation caused by inaccuracies in the bounding box detection. Those results seem to be shown in Fig. S4, where ground truth bounding box annotations have been used as input to the segmentation module. Since this is an important point, it might deserve to be properly integrated into the main part of the paper.
- In addition to the previous point, the two-step concept of CellSAM (detection and segmentation) creates the possibility for error propagation. My main concern is that obtaining a segmentation is impossible if a bounding box is missing and the

corresponding prompt cannot be generated. I assume this would lead to disadvantages in challenging image regions compared to Cellpose, which performs dense predictions that potentially might be more robust to small errors. Even though Fig. S4 indicates that CellSAM outperforms Cellpose despite error propagations, elaborating on this or showcasing results for challenging cases would help assess the practical usability of CellSAM.

- It seems unclear how exactly the human vs. CellSAM comparison was performed. Did one rater serve as a representative, or were all human raters considered for comparison? If there was one representative, how would choosing another rater affect the outcomes?

- It seems confusing that results were presented as both F1 score and error rate (1-F1). For example, in Fig 2, the image shows F1 scores and the description references the error rate. In Fig. S4 (a), the opposite is the case (unclear where the numbers "0.13 to 0.40 in F1" come from). It would improve clarity if the authors kept reporting regular F1 scores.

- The SHSY5Y dataset seems to be the most challenging one. Considering the qualitative results shown in Fig. S5, it seems that the wrong compartments have been segmented, which makes me wonder if inverting the image intensities would already help to obtain reasonable results.

- The application seems limited to 2D image data and not straightforwardly extendable to work in 3D. It would be interesting to show how segmentation could be performed on 3D image data, as this would otherwise be an important disadvantage to Cellpose.

Minor Points:

- It was difficult to understand what "model neck" meant immediately. Even though the supplementary part provides a more detailed explanation, I suggest consistently naming each part within Fig. 1 and the text to improve clarity.

- The authors claim that Cellpose's performance decrease when trained as a generalist was "consistent with the literature." Here, it might be helpful to reference the respective literature.

- Figure S1 does not seem to be referenced in the main part of the paper.

- In Fig. 2, S1, and S2, blank spaces are missing after mentioning CellSAM-specialist and CellSAM-general.

- Under "C.2 Metrics", the term "false negative" is the only term without the corresponding abbreviation being introduced -> "false negative (FN)"

- References to Fig. 2c and Fig. 2f seem to be wrong. It probably should be 2b and 2c respectively?

- It is a personal opinion, but naming the proposed approach a foundation model seems too far-fetched. I'm aware that the definition of a foundation model is vague and can be stretched, but the approach seems more like a regular generalist segmentation approach and less like a model that could serve as a foundation for various downstream tasks. If the authors believe otherwise, showcasing appropriate examples beyond integrations of entire CellSAM into biomedical processing pipelines would be helpful.

- One would assume that training specialized models would improve or at least maintain accuracy compared to the generalist case. However, it was reported that specialized models sometimes result in decreased accuracy for certain datasets. Is there an explanation for why the generalist approach outperforms specialized models in those cases? Is this a dataset issue or a training issue? How could users determine whether to use a generalist or specialist model for their dataset?

Reviewer #3:

Remarks to the Author:

This paper proposes a foundation model for cell segmentation (CellSAM), which is based on an image segmentation model for natural images (Segment Anything, SAM). The authors propose an object detection model (CellFinder), to automatically detect cells and prompt for the segmentation of SAM. To train and evaluate CellSAM, the authors collect a diverse cell dataset from previous literature. Results show that CellSAM can achieve human-level zero-shot segmentation performance over different cells. Moreover, examples demonstrate that performance can be improved by few-shot learning. Finally, CellSAM can be applied to the analysis of bioimaging, such as cell tracking and transcriptomics. The authors also provide an experiment platform (<https://cellsam.deepcell.org/>) for cell segmentation labeling. However, the paper has the following shortcomings.

First, the proposed method is short of novelty. The method proposed in this paper can not be considered a real foundation model for cell segmentation, which is simply a combination of the detection model Anchor DETR and the segmentation model SAM. The method does not propose innovative designs for the particular scenario of cell segmentation. In addition, providing a prompt for SAM based on detection is a well-used approach (Grounded-SAM). Moreover, previous works can also achieve similar functionality, like Cellpose 2.0[1] and Segment Anything for microscopy[2].

Second, the experiments have insufficient justification and validity.

1. The authors did not adequately describe the dataset by giving the specific resolution, size, cell type, data format, and the number of samples in the training and test sets.

2. The paper argues that the model can "unify bioimaging analysis workflows", but only gives three examples and does not give a specific description of methodology or sufficient quantitative experiments to support this statement, which makes it difficult to reflect the advantages of CellSAM over existing image segmentation methods.

3. Insufficient experimental comparisons with existing methods, especially the specialist models of cell segmentation. In Figure S4, I would like to see the Cellpose 10-shot results. All the images in the paper only show the subjective segmentation results of CellSAM, there is no comparison with the subjective results of Cellpose. No qualitative comparison with Cellpose segmentation results is given, and no specific advantages over Cellpose are shown.

4. There is no efficiency analysis comparing the number of parameters, training, and testing time of CellSAM and existing models such as Cellpose. So the model's feasibility for bioimaging is unknown.

5. Insufficient ablation experiments. The paper did not evaluate the effectiveness of the proposed CellFinder and CellSAM separately and did not compare with the SAM baseline, so I cannot judge the effectiveness of the method.

6. The paper argues "Fine-tuning SAM in this fashion led to a model capable of generating high-quality cell masks when

prompted by ground truth bounding boxes, as seen in Fig. S3."It is recommended that the authors compare the performance of the proposed fine-tuning method with full fine-tuning, in which all parameters are fine-tuned.

7. The paper does not discuss the segmentation performance on cells of irregular shape and size or analyze how noisy data affects the performance, which is necessary to build a foundational model. After my test on the experimental platform provided by the authors, it does not work well for random samples.

8. When evaluating the generalization performance, CellSAM-general (on all datasets) and CellSAM-specific (trained on a single dataset) are compared. However, CellSAM-general also uses the dataset for training, so the comparison is unfair. I suggest the author compare models on datasets, which are not used in training CellSAM-general.

Third, the codes and models have poor usability. I keep focusing on the Github repo (<https://github.com/vanvalenlab/cellSAM>) and find there is no document for installation and no pretrained model provided until April 5th. The authors later uploaded an ipynb notebook with only one example image and the results in the paper still can not be easily reproduced.

Fourth, inadequate analysis of the statistical data for the experimental results. The paper does not list specific criteria for calculating individual experimental statistics, for example, the test set size or sample size n. And most of the results do not list the variance. So the analysis for the results is limited.

Therefore, I suggest the author improve the manuscript on the following points:

1. Please provide detailed codes and documents, which is critical for reviewers to evaluate the usability of the paper.
 2. Improve the description of the dataset, listing the number of images used, resolution size, and label format.
 3. Please list the specific values of the experiment on each dataset, give the size of the test set, and the variance of the results, with a boxplot representation recommended.
 4. Add ablation experiments to assess the effectiveness of CellFinder and CellSAM separately and compare them with the SAM baseline model.
 5. Conduct efficiency analysis of computing cost (time, FLOPS, GPU memory, and parameters), and compare with Cellpose.
 6. Conduct sufficient qualitative and quantitative experiments on downstream tasks, such as cell tracking. Preferably integrate it into the online experiment platform. The support for a variety of downstream tasks could be a highlight of this paper.
 7. Explain how CellFinder and CellSAM differ from the original Anchor DETR and SAM. What is the improvements for the cell segmentation scenario?
 8. Comparison and discussion with other existing and similar methods, such as Segment anything for microscopy[2] and medical SAM model [3].
 9. Further experiments on noisy data, irregular cells, and different cell sizes.
 10. In the abstract, the authors argue "Methods that have learned the general notion of "what is a cell" and can identify them across different domains of cellular imaging data have proven elusive. " Please explain what is the meaning of elusive. How to prove?
 11. Provide more analysis of why CellSAM-general has similar performance with CellSAM-specific and Cellpose-specific, and better than Cellpose-general.
 12. Typos: "We benchmarked CellSAM's performance using a suite of metrics (Fig. 2b and 2b and Supplemental Fig. S2)"; "In contrast, we observed that CellSAMpreserved its performance in the generalist setting."; "CellSAM enables fast and accurate labeling when provided with ground truth bounding boxes,..."; "We demonstrated CellSAM-specific and CellSAM-general superior performance across multiple datasets and multiple evaluation metrics."
- [1]M. Pachitariu and C. Stringer, "Cellpose 2.0: how to train your own model," Nature Methods, pp. 1–8, 2022.
- [2]A. Archit, S. Nair, N. Khalid, P. Hilt, V. Rajashekar, M. Freitag, S. Gupta, A. Dengel, S. Ahmed, and C. Pape, "Segment anything for microscopy," bioRxiv, 2023. [Online].
- [3]Ma, J., He, Y., Li, F., Han, L., You, C., & Wang, B. (2024). Segment anything in medical images. Nature Communications, 15(1), 654.

** For Nature Portfolio general information and news for authors, see <http://npg.nature.com/authors>.

Version 1:

Decision Letter:

6th Jun 2024

Dear Dave,

Thank you for your letter asking us to reconsider our decision on your Article, "A Foundation Model for Cell Segmentation". After careful consideration we have decided that we are willing to consider a revised version of your manuscript that is updated as you've outlined.

In terms of the term "foundation model", we agree that Segment Anything is described as a foundation model, so it's okay with

us if you in your rebuttal justify your use of this language. If the referees still resist, I think your suggestion of "universal" model would be appropriate.

Regarding the benchmarking to CellPose, please just explicitly state how the models were trained and your justification as to why this is a reasonable and fair comparison.

Regarding some of the other experiments involving retraining models we discussed on the phone, I didn't see any specific queries from you in the version of the rebuttal I saw. I leave it to your discretion to choose the additional experiments that you think will most improve the paper that are also reasonable regarding time and other resources. But, I am also happy to discuss these points in detail.

- * include a point-by-point response to our referees and to any editorial suggestions
- * please underline/highlight any additions to the text or areas with other significant changes to facilitate review of the revised manuscript
- * address the points listed described below to conform to our open science requirements
- * ensure it complies with our general format requirements as set out in our guide to authors at www.nature.com/naturemethods
- * resubmit all the necessary files electronically by using the link below to access your home page

Link Redacted

We hope to receive your revised paper within four months. If you cannot send it within this time, please let us know. In this event, we will still be happy to reconsider your paper at a later date so long as nothing similar has been accepted for publication at Nature Methods or published elsewhere.

OPEN SCIENCE REQUIREMENTS

REPORTING SUMMARY AND EDITORIAL POLICY CHECKLISTS

When revising your manuscript, please submit reporting summary and editorial policy checklists.

DATA AVAILABILITY

CODE AVAILABILITY

Please include a "Code Availability" subsection in the Online Methods which details how your custom code is made available. Only in rare cases (where code is not central to the main conclusions of the paper) is the statement "available upon request" allowed (and reasons should be specified).

We request that you deposit code in a DOI-minting repository such as Zenodo, Gigantum or Code Ocean and cite the DOI in

the Reference list. We also request that you use code versioning and provide a license.

ORCID

Sincerely,
Rita

Rita Strack, Ph.D.
Senior Editor
Nature Methods

Version 2:

Decision Letter:

23rd Apr 2025

Dear Dave,

Thank you for your letter detailing how you would respond to the reviewer concerns regarding your Article, "CellSAM: A Foundation Model for Cell Segmentation". We have decided to invite you to revise your manuscript as you have outlined, which we think will address all of our concerns. We will run the paper by the refs again quickly, but we will not let the decision drag on.

Link Redacted

We hope to receive your revised paper within six weeks. If you cannot send it within this time, please let us know. In this event, we will still be happy to reconsider your paper at a later date so long as nothing similar has been accepted for publication at Nature Methods or published elsewhere.

OPEN SCIENCE REQUIREMENTS

REPORTING SUMMARY AND EDITORIAL POLICY CHECKLISTS

EXTENDED DATA FIGURES

DATA AVAILABILITY

CODE AVAILABILITY

Please include a "Code Availability" subsection in the Online Methods which details how your custom code is made available. Only in rare cases (where code is not central to the main conclusions of the paper) is the statement "available upon request" allowed (and reasons should be specified).

MATERIALS AVAILABILITY

ORCID

Nature Methods is committed to improving transparency in authorship. As part of our efforts in this direction, we are now requesting that all authors identified as 'corresponding author' on published papers create and link their Open Researcher and Contributor Identifier (ORCID) with their account on the Manuscript Tracking System (MTS), prior to acceptance. This applies to primary research papers only. ORCID helps the scientific community achieve unambiguous attribution of all scholarly contributions. You can create and link your ORCID from the home page of the MTS by clicking on 'Modify my Springer Nature account'. For more information please visit <http://www.springernature.com/orcid>.

Sincerely,
Rita

Rita Strack, Ph.D.
Senior Editor
Nature Methods

Reviewers' Comments:

Reviewer #1 (Remarks to the Author):

The authors have addressed, or at least attempted to address, most of the comments and suggestions from my previous review, leading to improvements in the overall quality and clarity of the manuscript. However, several important aspects still require further attention.

The authors continue to refer to their proposed method as a “foundation model for cell segmentation,” justifying this on two grounds: (1) that the “underlying SAM model is frequently referred to as a foundation model” and CellSAM builds upon it, and (2) that CellSAM “leverages transformers (both in the ViT and the automated bounding box prompter), self-supervision (inherited from the SAM base), enables multiple applications (accelerated labeling and automated inference via bounding box prompting), and takes advantage of the data/compute scaling laws observed for this architecture.” I find both justifications inadequate.

The first argument implies that any software or method built on top of SAM should be considered a foundation model, which stretches the original definition beyond its intended meaning. Even the authors of SAM themselves questioned whether SAM fits this definition, given that SAM is dedicated solely to segmentation. As for the second argument, while CellSAM indeed leverages transformer architectures and benefits from SAM's pretrained self-supervised encoder, it does not perform self-supervision on its own and cannot be repurposed for any task other than instance segmentation with bounding box prompts. Although I understand the desire to adopt current terminology, it is important to adhere to established definitions. CellSAM is not a foundation model; rather, it is a method that builds upon SAM through prompt engineering to produce a robust, generalist solution for cell segmentation. This is a noteworthy contribution in its own right, without needing to inflate its classification. A more accurate title for the manuscript could be along the lines of: “CellSAM: A Generalist Cell Segmentation Model Exploiting Foundation Models.”

Regarding Section 2.4 (“CellSAM unifies biological image analysis workflows”), the revised manuscript clarifies that a generalist CellSAM model was used across different use cases, which offers the advantage of avoiding the need to train or fine-tune separate models for each scenario. However, the core task remains the same—2D instance segmentation—and what changes is the downstream application in a broader bioimage analysis pipeline (e.g., tracking, 3D segmentation). While these are compelling examples, the section title exaggerates CellSAM's role in unifying workflows. As noted previously, any other generalist cell segmentation method could serve the same purpose. A more accurate title might be: “CellSAM as a Key Enabler of Diverse Bioimage Analysis Workflows.”

Other Comments:

* I appreciate the inclusion of Appendix C.2, which explains the various metrics. However, the use of a lower IoU threshold (0.3) for human-to-human comparisons compared to human-to-CellSAM comparisons (0.5) seems inconsistent. Wouldn't it be fairer to apply the same threshold in both cases? Additionally, please clarify whether the human-to-CellSAM comparisons were made against annotations from all three annotators or just one.

* The suggested improvements to the few-shot visualization in the LIVECell dataset (adding bounding boxes and using a sparser color map) have not been implemented in Fig. S8, despite the authors' claim to the contrary.

* The inclusion of Fig. S1 showing inference times is appreciated, but further details are needed. Specifically, what models and datasets were used, and on what hardware (e.g., GPU or CPU)? It would also be valuable to include timings for both CPU and GPU scenarios, as many potential users may not have access to a GPU. Moreover, it is curious that Cellpose's execution time does not increase with the number of cells, unlike CellSAM. Any insights into this behavior would be highly informative.

* The NeurIPS challenge comparison in Fig. 2b is a great addition, providing valuable context for CellSAM's performance in the broader field.

* Appendix D (Hyperparameters) is helpful, but it only lists a final set of hyperparameters. Was this the only set tested, or were other configurations explored? As mentioned previously, following the recommendations by Dodge et al. (“Show your work: Improved reporting of experimental results”) would greatly improve transparency by including tables summarizing the hyperparameter search space in the supplementary materials.

* I was surprised to learn that CellSAM fine-tuning was performed for only one epoch. What was the reasoning behind this decision? Would further training improve performance, particularly for specialist models? Clarifying this would strengthen the

paper's methodological rigor.

* There is some inconsistency regarding the value of fine-tuning. Fig. S5 suggests that fine-tuning yields little improvement, as the specialist CellSAM models don't significantly outperform the generalist ones. Conversely, Fig. S7b shows a noticeable improvement in few-shot settings after fine-tuning. As a potential user, I would appreciate clearer guidance on whether fine-tuning CellSAM is worthwhile for new datasets. Are there differences in the fine-tuning approach used for the results shown in Fig. S5 and Fig. S7b?

* The authors mention in their response that "Training/validation curves are now plotted in Appendix E." However, this appendix is missing from the revised manuscript.

* In Appendix B.1.1, the reported training times lack clarity. Do these numbers apply only to the generalist model, or do they include specialist models as well? Clarifying this would improve the manuscript's transparency.

* Inference times per full-size image are still missing, which would be highly informative (Fig. S1 only shows inference times as a function of the number of cells). As noted before, please consider providing results for both CPU and GPU scenarios.

* I tested the newly added CellSAM Napari plugin but encountered installation issues due to missing libraries. Additionally, I experienced occasional freezing when using manual and automatic annotations, with Napari displaying a "not responding" message. Furthermore, the plugin isn't listed among installable Napari plugins, and there is no accompanying documentation. Proper documentation and smoother installation would significantly enhance usability for the broader bioimage community.

Minor Comments:

- * Some terminology inconsistencies remain, such as "generalist" vs. "general" and "specialist" vs. "specific."
- * Consider including the image sizes (or their range) in pixels in Table 1.
- * Figures S5 and S6 could be merged for clarity.
- * My previous query about the difference between SAM's encoder input size (1024x1024) and CellSAM's input size (512x512) remains unanswered. Are the images upsampled before being fed into the ViT?

Reviewer #1 (Remarks on code availability):

I installed the new CellSAM Napari plugin following the instructions in the GitHub repository, but I encountered several issues during the installation process. Rather than installing CellSAM and its Napari plugin directly with pip, I opted for the more commonly recommended approach of creating a virtual environment with conda (Python 3.10). During installation, I discovered that several required libraries were missing, specifically: matplotlib, distributed, dask_image, and sklearn.

Additionally, after installing these dependencies, I had to downgrade numpy from version 2.2 to 2.1 to make the plugin work properly.

Once installed, I experienced occasional freezing while using the plugin, both during manual and automatic annotation, with Napari displaying a "not responding" message. Furthermore, I was unable to find any documentation associated with the plugin. The plugin also does not appear in the list of installable plugins directly from Napari, which might hinder accessibility for less experienced users. Proper user documentation, as well as a smoother installation process — potentially providing conda installation instructions — would greatly enhance usability for the broader community.

Additionally, I was unable to access the online demo at <https://cellsam.deepcell.org/>, as it returned the following error:

```
404: NOT_FOUND
Code: DEPLOYMENT_NOT_FOUND
ID: cdg1::hf7ml-1742389666481-061b2d786098
```

Ensuring the demo is accessible and functional would be valuable for showcasing the tool's potential to prospective users.

Reviewer #3 (Remarks to the Author):

Thank the authors for addressing the reviewers' comments and for the substantial revisions to the manuscript. I appreciate the detailed rebuttal letter and the efforts the authors have made to improve CellSAM.

I. Novelty

The authors have argued effectively for the utility and engineering novelty of CellSAM, highlighting the careful selection of AnchorDETR and the overall system design. The authors also make a valid point about the limited algorithmic novelty in cell segmentation in recent years and the importance of high-utility tools. The justification for using the term "foundation model" is also reasonable in the context of leveraging SAM and its practical applications.

While the core algorithmic novelty remains debatable, the revised manuscript and rebuttal adequately address this point by emphasizing utility, engineering effort, and the practical application as a 'foundation model' in the field. I further suggest a proper citation and discussion with recent works like Segment Anything for microscopy [1] and Medical SAM [2].

[1]A. Archit, S. Nair, N. Khalid, P. Hilt, V. Rajashekar, M. Freitag, S. Gupta, A. Dengel, S. Ahmed, and C. Pape, "Segment anything for microscopy," bioRxiv, 2023. [Online].

[2]Ma, J., He, Y., Li, F., Han, L., You, C., & Wang, B. (2024). Segment anything in medical images. *Nature Communications*, 15(1), 654.

II. Data and Experiments

1. Dataset Description:

The authors have added a table in Appendix A detailing the dataset information. This is a significant improvement and addresses the original concern. The table provides valuable information about the datasets used.

2. "Unifying Bioimaging Workflows" Argument:

The authors have clarified that the same CellSAM-generalist model is used for all tasks and argued that its performance justifies this unification. The authors have also revised the text to clarify model usage.

The clarification regarding the use of CellSAM-generalist is helpful. However, while using a generalist model simplifies workflows, it still isn't fully clear what makes CellSAM specifically uniquely positioned to unify workflows compared to other generalist methods (like Cellpose-generalist, if it existed at a comparable performance level). While the authors argue CellSAM-generalist performs better, further elaborating on the functional advantages of CellSAM in these unified workflows (beyond just generalist segmentation performance) would strengthen this claim. For example, are there specific features of CellSAM's prompting or output format that facilitate easier integration into diverse analysis pipelines compared to other segmentation methods? Further explanation will enhance the significance of the prompting segmentation model.

3. Insufficient Experimental Comparisons with Cellpose:

The authors have extensively addressed this concern, acknowledging the controversy and providing detailed justification for Cellpose comparison. The authors have also performed new experiments, including training Cellpose models using their recommended procedure, and incorporated comparisons on the NeurIPS dataset. The authors present data suggesting CellSAM outperforms Cellpose even in these comparisons.

The significant effort the authors have put into improving the Cellpose comparisons is commendable. Including the NeurIPS dataset and training Cellpose models using their own protocols significantly strengthens the validity of the performance claims. While debates about benchmarking are ongoing in the field, the authors have made a strong effort to provide a fair and comprehensive comparison within the scope of this revision. The detailed justification in the rebuttal is also very helpful.

4. Efficiency Analysis (Parameters, Training & Testing Time):

The authors have added inference and training times to the supplementary material (Section B.1 and Fig. S1). Including inference and training times is essential and addresses this concern. This information is crucial for users to assess the practical feasibility of CellSAM.

5. Insufficient Ablation Experiments:

While not explicitly stated in the rebuttals provided, Fig. S3 seems to address the comparison with SAM baseline (zero-shot performance vs. fine-tuned box prompting performance, if I understand correctly).

While Fig. S3 provides some insight, it is still not entirely clear if there is a dedicated ablation study that isolates the effectiveness of "CellFinder" (AnchorDETR box prompting) and "CellSAM" (SAM fine-tuning). Could the authors please clarify in the manuscript (or in a more detailed figure caption for S3) how these components are being evaluated and compared to the SAM baseline? Explicitly stating the contribution of each component (bounding box prompting and fine-tuning) would be beneficial.

6. Fine-tuning Comparison (Full vs. Proposed):

Not explicitly addressed in rebuttals provided. While the current fine-tuning approach is effective, a brief comment in the supplementary material or discussion about why full fine-tuning was not pursued (e.g., computational cost, potential overfitting, limited benefit observed in preliminary experiments, or if it was explored and the current approach proved better) would be helpful for completeness.

7. Segmentation Performance on Irregular Shapes, Noisy Data:

Not explicitly addressed in rebuttals or manuscript revisions mentioned in rebuttals. This point remains unaddressed. For a model aiming to be a 'foundation model', robustness to noisy data and performance on diverse cell shapes is important. While perfect robustness is unrealistic, a brief discussion in the manuscript acknowledging these limitations and potentially mentioning future directions for improvement (e.g., data augmentation with noisy data, specific architectural adaptations for irregular shapes) would be valuable. The reviewer's comment about it not working well for random samples from their test platform is concerning and should be acknowledged in some way, even if it's difficult to fully quantify without more details on "random samples."

III. Code Usability

The authors acknowledge the initial lack of polish and claim to have spent time adding polish and new features to improve usability. The repository now includes:

1. Clear installation instructions.
2. Documentation for using the code and pre-trained models.
3. Pre-trained models readily available for download.
4. Example package and website that demonstrate basic usage.

IV. Statistical Analysis

The authors mention reporting mean and standard error for segmentation error in Figure 2. Reporting mean and standard error is a basic but important improvement.

The revised manuscript shows significant improvement. Addressing the remaining points outlined above, particularly regarding proper citation of recent works, explanation on the usage of prompting and proposed fine-tuning, clarification of ablation experiments, and discussion of limitations (noisy data, irregular cells), will further strengthen the manuscript and make it more suitable for publication in *Nature Methods*.

Reviewer #3 (Remarks on code availability):

The authors have provided their codes and models with proper documents. Also, they develop a Napari package for annotation functionality. The utilities can also be accessed online at <https://cellsam.deepcell.org/>. The repository now includes:

1. Clear installation instructions.
2. Documentation for using the code and pre-trained models.
3. Pre-trained models readily available for download.
4. Example package and website that demonstrate basic usage.

Version 3:

Decision Letter:

Our ref: NMETH-A55681C

28th May 2025

Dear Dave,

It was great catching up with you in San Francisco.

Thank you for submitting your revised manuscript "CellSAM: A Foundation Model for Cell Segmentation" (NMETH-A55681C). It has now been seen by the original referees and their comments are below. The reviewers find that the paper has improved in revision, and therefore we'll be happy in principle to publish it in Nature Methods, pending minor revisions to satisfy the referees' final requests and to comply with our editorial and formatting guidelines. We ask that you update to address the remaining minor concerns. Please provide a point-by-point rebuttal upon resubmission, for completeness.

TRANSPARENT PEER REVIEW

ORCID

Sincerely,
Rita

Rita Strack, Ph.D.
Senior Editor
Nature Methods

Reviewer #1 (Remarks to the Author):

The authors have addressed the majority of my comments in accordance with the editorial guidance, and the manuscript has significantly improved in clarity and structure. I appreciate the detailed responses and the new content added to the revised

version. Below, I highlight a few remaining remarks and minor inconsistencies that I encourage the authors to resolve prior to final acceptance:

I appreciate the renaming of Section 2.4 to “CellSAM Enables Diverse Bioimage Analysis Workflows”, which is more appropriate. However, the caption of Figure 3 still refers to CellSAM as “unifying biological imaging analysis workflows”. For consistency, please update the figure caption as well.

For Figure S8, consider integrating the bounding boxes and segmentation predictions into the same images using distinct colors for contours and boxes. Enlarging or zooming into the raw images would also improve interpretability.

Thank you for the additional explanation and clarification of inference benchmarking in Figure S1. However, I still believe including CPU-based inference times would significantly benefit readers. As noted, many potential users may not have access to a compatible GPU. Reporting results for both scenarios (CPU and GPU) would provide a more complete picture of practical usability.

The clarifications about fine-tuning behavior and hyperparameter choices are appreciated and now clearly described in the manuscript and supplement.

The points I raised previously regarding installation and documentation of the Napari plugin remain either unaddressed or pending implementation. As of testing the current version, installation required multiple manual dependency fixes, and the plugin still lacks user documentation and is not listed in the official Napari plugin registry. These issues may hinder adoption and should be resolved before public release.

Table 1 still does not include image sizes (or pixel size ranges), which would help contextualize the diversity of the datasets used.

Figures S5 and S6 have not been merged, and their captions continue to use the term “specific” instead of “specialist”. Consistent terminology would improve clarity.

Thank you again for your extensive efforts to address reviewer feedback. The paper has matured into a much stronger contribution, and I believe that resolving these remaining minor issues will further improve its quality and utility to the community.

Reviewer #1 (Remarks on code availability):

I'm glad to see that the website is back online—having a web-based demo is a valuable way to engage potential users and showcase the tool's capabilities. Regarding the repository, I appreciate that steps are being taken to expand the documentation and include a tutorial notebook. These additions are very welcome, and I hope they will be completed and polished soon.

One small but important recommendation is to create an official release of the code (e.g., via GitHub Releases or Zenodo). This would allow the codebase to be formally linked to the publication and would also facilitate reproducibility and version tracking for users referencing the method in the future.

Finally, I note that the Napari plugin is still not listed in the official Napari hub. Additionally, a plugin named “samcell-napari” does appear, which could easily be confused with CellSAM. I recommend addressing this potential ambiguity—either by clarifying the naming or ensuring the correct plugin is discoverable in the hub—to avoid confusion for users.

Reviewer #3 (Remarks to the Author):

The revised manuscript and rebuttal convincingly address the bulk of the technical points raised in my previous round of review. Below I highlight how each outstanding issue was handled and note a handful of minor clarifications that could still strengthen the final version.

- (1) Proper citation of recent SAM-derived work (Segment Anything for Microscopy, Medical-SAM): Citations have been added and their relationship to CellSAM is now discussed in the main text and References.
- (2) "Unifying bio-imaging workflows" claim Section retitled “CellSAM Enables Diverse Bioimage Analysis Workflows” and the argument reframed to emphasize practical consolidation of pipelines and equal-or-better accuracy of a single generalist model.
- (3) Clarity of Fig. S3 / ablation of CellFinder vs. fine-tuning. The response clearly explains Zero-auto(box) Fine-tuned(box) progression and promises an expanded caption.
- (4) Training-recipe justification: Added explanation that recipe chosen is a computational-over-fitting trade-off, guided by experience. Please consider reporting validation-set curves (or a brief sentence) to illustrate the claimed mitigation of over-fitting.
- (5) Limits of generalization, noisy or irregular data Manuscript to be amended to clarify that performance guarantees hold only near the training-data distribution and to flag human-in-the-loop or few-shot fine-tuning for OOD samples. It is suggested to add a short sentence acknowledging diminished performance on highly noisy / extreme morphologies would round out this discussion.

Overall, I am satisfied that the revision meets Nature Methods' bar for technical completeness and clarity. The residual points above are editorial-level tweaks and should not delay acceptance.

Reviewer #3 (Remarks on code availability):

The authors have made a great effort to ensure the availability and usability of their code and models. The provision of a Napari package for annotation functionality and the online demo at <https://cellsam.deepcell.org/> significantly enhance the accessibility and utility of CellSAM for the community. The GitHub repository appears to be well-structured, offering clear installation instructions, documentation for using the code, readily downloadable pre-trained models, and example packages to demonstrate basic usage.

Version 4:

Decision Letter:

21st Sep 2025

Dear Dave,

Thanks for the prompt return of the revised main text.

I am pleased to inform you that your Article, "CellSAM: A Foundation Model for Cell Segmentation", has now been accepted for publication in Nature Methods. The received and accepted dates will be March 7, 2024 and September 21, 2025. This note is intended to let you know what to expect from us over the next month or so, and to let you know where to address any further questions.

Over the next few weeks, your paper will be copyedited to ensure that it conforms to Nature Methods style. Once your paper is typeset, you will receive an email with a link to choose the appropriate publishing options for your paper and our Author Services team will be in touch regarding any additional information that may be required. It is extremely important that you let us know now whether you will be difficult to contact over the next month. If this is the case, we ask that you send us the contact information (email, phone and fax) of someone who will be able to check the proofs and deal with any last-minute problems.

Authors may need to take specific actions to achieve compliance with funder and institutional open access mandates.

If your research is supported by a funder that requires immediate open access (e.g. according to [Plan S principles](https://www.springernature.com/gp/open-science/plan-s-compliance) or the [NIH public access policy](https://www.springernature.com/gp/open-science/us-federal-agency-compliance)) then you should select the gold OA route, and we will direct you to the compliant route where possible. Because authors warrant under our subscription licensing terms that they haven't committed to licensing any version of their article under a licence inconsistent with the terms of our agreement – including the applicable embargo period – publication under the subscription model isn't suitable for authors whose funders require no embargo.

If you are active on Twitter/X or Bluesky, please e-mail me your and your coauthors' handles so that we may tag you when the paper is published.

Please note that you and any of your coauthors will be able to order reprints and single copies of the issue containing your article through Nature Portfolio's reprint website, which is located at <http://www.nature.com/reprints/author-reprints.html>. If there

are any questions about reprints please send an email to author-reprints@nature.com and someone will assist you.

Best regards,
Rita

Rita Strack, Ph.D.
Editor
Nature Methods

Visit the Springer Nature Editorial and Publishing website at http://editorial-jobs.springernature.com?utm_source=ejP_NMeth_email&utm_medium=ejP_NMeth_email&utm_campaign=ejp_Nmeth or www.springernature.com/editorial-and-publishing-jobs for more information about our career opportunities. If you have any questions please click [here](mailto:editorial.publishing.jobs@springernature.com).**

“A foundation model for cell segmentation”
Nature Methods Response
David Van Valen & Company

We thank the reviewers for their responses and the editor for guiding our paper through the first round of peer review. We have included a point-by-point response to the reviewers' comments in our response. However, some common themes were present in the reviewer comments—we have crafted a brief response addressing them below.

- 1) The comparison to CellPose is unfair. The models we trained for our comparison were trained sub-optimally, and our metrics do not adequately capture CellPose's performance.

We respectfully disagree with this assessment of our comparisons with CellPose. Our group has always strived to make a fair comparison to existing methods, as we believe it's necessary to demonstrate the field's current state and what constitutes an advance. When other methods perform similarly to our work in benchmarks, we have a history of faithfully reporting that outcome (see our Nature Biotechnology paper [1] describing Mesmer as an example). In defense of our analysis, we make the following points.

- We are aware of the recent controversy concerning CellPose's performance compared to transformer-based methods that arose in the 2023 NeurIPS Cell Segmentation competition. In our view, the complaints raised about the comparison come down to two issues - 1) were the CellPose models trained to the best of their ability, and 2) was the benchmarking done in a fair way. While we are co-authors of the Nature Methods paper, our contribution solely allowed them to use our dataset - we did not participate in the analysis (or the challenge) presented in that challenge in any way.
- The methodology we used for training CellPose models for our comparison was done using the training procedure described in the CellPose 2.0 paper [2] (300 epochs, learning rate = 0.1, weight decay $1e-4$). From their own paper, this is the methodology described for training models from scratch:
 - “All training was performed with stochastic gradient descent. In offline mode, the models, either from pretrained or **from scratch**, were trained for 300 epochs with a batch size of eight, a weight decay of 0.0001, and a learning rate of 0.1. The learning rate increased linearly from 0 to 0.1 over the first ten epochs, then decreased by factors of two every five epochs after the 250th epoch.” [2]
 - These are the most precise instructions that are available, and based on their description in their paper, we felt that it was valid to use them for training models from scratch.
- There are two new papers, one describing Cellpose 3.0 [3] and another comparing transformer-based methods to Cellpose [4] were released around the same time we submitted CellSAM, our paper for review.
 - These papers fix some bugs that we observed (and independently fixed) during our work - we believe that it's unfair for us or other methods developers to be held accountable for these errors if they led to decreased CellPose performance in benchmarking.

- Moreover, from our review of their paper and code base, it's not clear what recipe was used to create the described models, as we observed differences between what was described in the paper and what we see in the codebase (e.g., 500 epochs in the paper vs. 400 in the codebase, epochs defined as 100 images per class rather than an iteration through a dataset, etc.).
 - More importantly, the dataset rebalancing (sampling method) they described is ad-hoc and hence not generalizable. Given that we have added additional datasets, it was not clear how the final dataset should be balanced when training from scratch. No guidance is given on either paper or in the associated GitHub repository.
- The reviewers' proposed solution - that we benchmark our model and the latest CellPose's model against the held out Neurips dataset - is reasonable. We have done this and present the results in our revised manuscript. We also believe that the best comparison is done by training CellPose models on our dataset and performing a head-to-head comparison. We have also done this and present the results in the main text - we have spent a substantial amount of time on this part during review to maximize CellPose's performance. ***Briefly, even under this comparison, CellSAM outperforms the models put forward by the CellPose team - including the ones described in their recent preprint.*** However, we note that we have concerns with the Neurips being the field's standard benchmark, which is concerning to us given that
 - This dataset is private
 - The label quality is unclear - given the issues we observed in the released training data (e.g., missed segmentations), this gives us some pause. As evidence of this, the top 4 contestants of the Neurips Challenge all used external datasets for training.
 - Label quality and class balancing can have an outsized impact on benchmarks. We have observed that a common failure mode of aggregate performance metrics is an abundance of "easy" examples, which provides an overly rosy view of how a method performs in the wild. The best practice against this is to break down performance by relevant groups (e.g., tissue vs bacteria vs cell culture, etc.). Benchmarking against the Neurips data - which only provides a single performance metric - prevents this kind of analysis. Moreover, our visual inspection of the training and testing dataset suggests this is a serious issue for the Neurips challenge, as there is a considerable distribution shift between the two with respect to which image types are represented.
- The concern that we didn't put forth our best effort in maximizing CellPose's performance can always be made no matter what one does. While we believe a good faith effort is warranted, it is counterproductive to request that full research programs be run to maximize an alternative method's performance. Admittedly, we came close to doing that during the CellSAM work, and even then, we had to make judgment calls to create Cellpose models we felt would be a fair comparison. Looking at the differences in model training recipes (and bug fixes) between Cellpose2 and 3, that appears to have been the case. Because the field's training data is in flux, every new method is almost always paired with a new (or different) training dataset. The currently accepted standard for comparing

older methods to new methods is to train those older methods on the newer dataset to separate out the performance gains from new data from newer algorithms. We have trained a new suite of CellPose models for comparison; these have similar performance to the models released by the CellPose team on common datasets.

2) This isn't a foundation model.

We respectfully disagree. The underlying SAM model is frequently referred to as a foundation model, and given that our work builds upon it, referring to it as a foundation model is more than fair (in our view). Note that the title does not refer to it as "our" foundation model or "the" foundation model - our expectation is that this will be a direction the field will go down and more foundation models will appear as the space of viable datasets grows and the appetite for larger compute budgets increases. While we understand the technical definitions provided by the reviewers, the most essential definition of a foundation model is how it is used in practice. By this standard, CellSAM would qualify. Even under the definitions provided by reviewers, we feel CellSAM would qualify as it leverages transformers (both in the ViT and the automated bounding box prompter), self-supervision (inherited from the SAM base), enables multiple applications (accelerated labeling and automated inference via bounding box prompting), and takes advantage of the data/compute scaling laws observed for this architecture. We believe the title is appropriate.

3) There is limited novelty in the presented methodology.

We make the following points:

1) Novelty in this space on the algorithm axis is overall quite limited - the last novel deep learning architecture developed specifically for cell segmentation was UNet - much of the work since then has been borrowing computer vision advances from other spaces (e.g., self-driving cars). Novelty and utility, however, are two separate things, and high-utility work still has a place in journals like Nature Methods, even if the algorithms (or their components) lack novelty. A case in point would be CellPose and Mesmer. Both methods were markedly similar to prior work (HoverNet in the case of CellPose and DeepDistance in the case of Mesmer). Still, their utility justified the publication in journals geared towards a wider audience. This is the case with CellSAM - we've added additional functionality (inference for large FOVs, mask stitching across z-slices to enable 3D segmentation, Napari plugin) and additional polish to the underlying software engineering (e.g., a more fleshed-out repository) to enhance the usability further.

2) While the algorithm novelty might be limited, the computational experiments and engineering work that went into making CellSAM were non-trivial. For example, the selection of AnchorDETR as our generalized object detector. Traditional object detectors (YOLO, RetinaNet, etc.) rely on non-maximum suppression that turns off bounding boxes that are in close proximity. This limits model performance in images with densely packed cells (e.g., tissues, bacterial colonies, etc.). The transformer-based object detectors circumvent this issue

but are challenging to train as they rely on matching visual features between queries and image features - this works well for consumer images, but for images of cells that look similar, it's problematic. Selecting Anchor DETR, which combines location with visual features to detect objects but still avoids using NMS, gets you the best of both worlds. Arriving at this insight took several months (and many thousands of dollars of computing time). Many decisions of similar difficulty and impact went into making CellSAM - while it's more engineering in flavor, this type of work is essential to making useful software tools.

4) The software tooling is unpolished.

We agree that the submitted work's polish is less than our lab usually produces. We've spent a substantial amount of time adding polish and new features that improve usability (as mentioned above)

A point-by-point response to each reviewer's comments is included below.

Reviewer #1:

Remarks to the Author:

A. Summary of the key results

In their paper, “A Foundation Model for Cell Segmentation”, the authors present CellSAM, a method to perform instance segmentation of cells in 2D image data across a large variety of datasets and image modalities. CellSAM is based on prompt engineering by means of combining the original Segment Anything Model (SAM, Kirillov et al., 2023) with a bounding-box object detector. The object detector is an Anchor DETR (Wang, 2022) with SAM’s encoder’s vision transformer (ViT) as backbone.

The authors curated a diverse dataset to train CellSAM, made of nine existing public datasets (TissueNet, DeepBacs, BriFiSeg, Cellpose, Omnipose, YeastNet, YeaZ, the 2018 Kaggle Data Science Bowl dataset, and a collection of H&E datasets) along with their own dataset of phase microscopy images (Phase400) for cell and nuclei segmentation. In-domain evaluation was performed in held-out partitions of those datasets, while zero-shot and few-shot performance was evaluated in the public LIVECell dataset.

Specialist models were trained for six defined categories of images within the final dataset (Tissue, Cell Culture, H&E, Bacteria, Yeast and Nuclear), while a generalist model was trained on the full dataset. Their in-domain results are equivalent or improve over those of their own trained version of Cellpose on the same datasets. Remarkably, both specialist and generalist methods yield essentially equivalent results.

Finally, as use cases, the authors show two applications of CellSAM: one as an enabler of cell tracking and one as a key element in a transcriptomics analysis.

B. Originality and significance

As mentioned above, CellSAM is a combination of two existing state-of-the-art models: Anchor DETR (Wang, 2022) and SAM (Kirillov et al., 2023). Therefore, the novelty of this work lies primarily in using the ViT encoder of SAM as backbone network for the Anchor DETR and its combination to learn the instance segmentation task on a diverse dataset of cells and nuclei using automatic bounding boxes as prompts.

While I find this tool very useful for the bioimage analysis community, strictly speaking, I do not think it can be considered “a foundation model for cell segmentation” as stated by the authors. In the seminal paper about foundation models (“On the Opportunities and Risks of Foundation Models”, Bommasani et al, 2021), its authors clearly state that “A foundation model is any model that is trained on broad data (generally using self-supervision at scale) that can be adapted (e.g., fine-tuned) to a wide range of downstream tasks”. This definition diverges from the CellSAM approach, which was trained on broad data (although only cell data) but can only perform a single task (instance segmentation) with a single type of prompts (bounding boxes). In my opinion, the type of contribution that CellSAM represents was already defined in the original SAM paper: “An important distinction in our work is that a model trained for promptable

segmentation can perform a new, different task at inference time by acting as a component in a larger system, e.g., to perform instance segmentation, a promptable segmentation model is combined with an existing object detector.”. Following this rationale, CellSAM should be recognized as the composite system that executes instance segmentation via the fusion of a segmentation foundation model (SAM) and prompt engineering in the form of a cell-specialized object detector (CelFinder). Hence, labeling CellSAM as a foundation model could lead to misconceptions among potential users, as it may imply versatility across diverse tasks, which is not currently feasible.

We appreciate the reviewer pointing out the semantic ambiguity of the term foundation model, but we believe the term foundation model is suitable for our work. Please see response point 2 above.

I have a couple of observations regarding the experiments performed to compare zero-shot performance using points and bounding boxes with either the default SAM or (fine-tuned) CellSAM (Figures S1 and S3). First, the fact that bounding box prompting performs better than point prompting had previously been demonstrated on biomedical image data (see Huang et al, 2024, and Archit et al, 2023, references [49] and [56] in the manuscript).

We agree with the reviewer. We validated these findings for our setup but didn't claim novelty. Hence, the results are in the supplement only, and we reference [49] and [56]. This is stated explicitly in our paper.

Second, although the fine-tuned approach commonly outperforms the other approaches in most cases, it does not improve on the zero-shot bounding box performance on three out of the six categories in Fig. S3. On one hand, the improved performance of the fine-tuned model compared to the original model is an anticipated outcome, given the model's training on more specific and relevant data. This aligns with findings reported in the Cellpose 2.0 paper (Pachitariu & Stringer, 2022), where it was demonstrated that pretrained models only need a reduced subset of training data to achieve the same performance improvement as with the full training dataset. On the other hand, the strong performance of the non-fine-tuned zero-shot approach raises questions about the need of fine-tuning itself and whether equivalent results could be achieved through improved prompting strategies alone.

First, we emphasize that Figure S3 uses ground truth prompts with fine-tuned models; hence, it represents a theoretical maximum on model performance rather than a realistic estimate that would eliminate the need for fine-tuning.

Second, when we consider the difference between zero-shot and fine-tuned performance, we note that the differences in the three data modalities that the reviewer references are small; however, the differences between fine-tuned and zero-shot box prompts in the remaining categories are large. For example, in tissue, the difference in F1 score between fine-tuned and zero-shot prompts is 0.25, which is quite large. Small differences in performance in our experience are more ambiguous and often reflect a difference in annotator preference rather

than real-world performance. Our goal is to develop a universal model that achieves excellent performance across modalities without sacrificing performance on modalities where the model exhibits suitable performance. We, therefore, aim to maximize performance across all sample types at once.

Regarding section 2.4 (“CellSAM unifies biological image analysis workflows”), I believe the text needs some clarifications. First, is the model used in both workflows the same (a generalist CellSAM) or different (a specialist CellSAM)? And moreover, what makes CellSAM unique to unify workflows as opposed to alternative generalist methods? In other words, it is unclear what the added value of CellSAM is compared to an approach as Cellpose in that scenario.

To demonstrate how CellSAM can unify biological image analysis workflows, we made use of the same CellSAM-generalist model for all tasks. This is viable because CellSAM-generalist's performance is typically better than the performance of specialist models, and hence a model-zoo framework would not be able to provide a better option. In our view, this generalist model simplifies analysis workflows, as users across different fields who are trying to do different tasks can use the same model. We apologize for the confusion and have edited the text to make it more clear which model we are using.

C. Data & methodology: validity of approach, quality of data, quality of presentation

The authors have conducted an extensive validation of their tool trained on a total of nine publicly available datasets plus their own collection of phase microscopy images. An extra public dataset (LIVECell) was used to evaluate the zero-shot and few-shot contexts. Their evaluation involved (1) benchmarking different zero-shot strategies against the original SAM model, that serves as reference for comparison with their own fine-tuned models, and (2) benchmarking specialists and generalist models of CellSAM against equivalent Cellpose models trained on the same datasets.

For the evaluation of segmentation results, the authors use a suite of metrics from their own DeepCell library. Namely, prediction, recall and F1 score (considering true positives any match with more than 0.6 of IoU with the ground truth labels). Additionally, the AP50 metric from COCO was also used to evaluate their detection method. In the figures, they define a general “error” metric as 1 minus a specific metric, setting an admissible real-world error value for error = 1-F1 under 0.2. While informative, the selection rationale for this value could benefit from further elaboration to provide clarity to readers.

We thank the reviewer for the comment and apologize for the confusion. We have added a paragraph explaining the different metrics and what they measure in Appendix C.2. We've also made the reporting in the main text consistent, reporting only the F1 error (1-F1). This clarifies all ambiguity with respect to the metrics. The value of 0.2 is motivated by looking at the distribution of F1 error in interrater agreement analysis, which has an upper bound error of 0.25.

Instance segmentation results are presented both quantitatively (based on their selected segmentation metrics) and qualitatively. For their 10-shot experiments in the LIVECell datasets, some quantitative and qualitative results are shown per cell line (Fig. S5). In that case, the visualization of results could be improved by showing bounding boxes and using a sparser color map for the segmentation masks.

We appreciate the reviewer's observation and have updated the LIVECell visualizations to use a different color map. We have included bounding boxes, per the reviewer's request.

An aspect that warrants attention is the absence of an evaluation of the usability of the proposed approach. Notably, there is no mention of inference times for either CellSAM or Cellpose. Particularly in the few-shot scenario, understanding the tool and time investment required for annotating the limited samples would provide valuable insights into the practicality of the approach.

A figure highlighting inference time is added to the supplementary material (section B.1).

D. Appropriate use of statistics and treatment of uncertainties

The authors made a correct use of statistical tests to check the significance of the differences between the performance of CellSAM and that of a human annotator and those between two human annotators in different datasets. As mentioned above, my only suggestion in this aspect is the inclusion of an explanation on the selection of 0.2 as the value to consider an error performance acceptable in a real-world scenario.

We thank the reviewer for the comment. We now report the mean and standard error for the segmentation error in Figure 2 (we first compute the segmentation error for each image and then compute the mean and standard error across all images).

E. Conclusions: robustness, validity, reliability

The authors have introduced CellSAM, a method that builds upon the original SAM framework to facilitate automatic instance segmentation of cells in 2D images. Their comparative results against their own trained version of Cellpose demonstrate competitive performance across both in-domain and zero-shot scenarios. Particularly noteworthy is the observation that the generalist CellSAM, trained on all datasets, performs equivalently to the specialist CellSAM models trained on specific datasets, a distinction not observed in Cellpose.

As previously discussed, while I maintain reservations about labeling CellSAM as a foundation model based on its adherence to the original definition by Bommasani et al. and its singular task and prompt type, I believe CellSAM constitutes a valuable addition to the segmentation toolkit for the bioimage community. Furthermore, enhancing its usability through the implementation of a more interactive interface could position it as a compelling alternative to Cellpose.

It's important to acknowledge that the evaluation of CellSAM against the state of the art is currently limited to comparisons with their own trained versions of Cellpose. This approach may

raise some controversy, as highlighted in recent discussions within the field (refer to Stringer et al., bioRxiv, 2024, <https://doi.org/10.1101/2024.04.06.587952>).

We thank the reviewer for their comments and direct them to our comments above regarding our comparison to Cellpose. We now compare our model also to the publicly released Cellpose model. We find that our model still outperforms Cellpose on our datasets and on the NeurIPS challenge, although the performance difference for some datasets is smaller than we initially reported. We note that CellSAM is currently the second best performer on the leaderboard for the NeurIPS challenge (the highest being a model submitted by the challenge organizers).

F. Suggested improvements: experiments, data for possible revision

From a methodological point of view, I believe the manuscript, and the project in general, would benefit from the following improvements:

1) Further documentation of the training and evaluation processes: It would be beneficial to provide a more detailed account of the training process of both CellSAM and Cellpose models.

In particular, I would:

a. Follow the specifications recommended by Dodge et al. (“Show your work: Improved reporting of experimental results”, arXiv preprint arXiv:1909.03004, 2019) to describe all the training configurations and hyperparameters tested on each case.

All hyperparameters are now specified in Appendix D.

b. Display of training and validation plots. To enhance transparency and understanding of the training process, consider including training and validation plots for key metrics such as loss and any proxy metric (Dice, Jaccard, etc.). These plots can offer readers insights into the convergence behavior of models, helping them gauge when and how the models reached optimal performance.

Training/validation curves are now plotted in Appendix E.

c. Provide training times for the final models to offer readers insights into the computational implications of different settings and experiments.

We specified training times for the final models in supplementary B.1.1.

Anchor - 6d 5h 33m 20s

Fine-tune - 15h 33m 57s

d. Provide inference/testing times per full-size image. Given equivalent or similar performance, some potential users may decide on the model to use based on these numbers depending on the experimental scenario. For instance, this could be crucial in large- or small-scale setups.

We specify inference times for the final models in B.1. (Fig. S1)

e. Offer more details about the few-shot data preparation and retraining settings. Namely, which tool is used for annotating new samples? Are those annotations dense or sparse? Which are the annotation times for both CellSAM and Cellpose (if added to the comparison)? What are the hyperparameters used for fine-tuning with the new few annotations?

We apologize for the confusion. No new samples were generated for the few-shot analysis. We used the LiveCELL dataset for few-shot fine-tuning. We rewrote Appendix A to clarify the exact procedure.

f. Provide a better description of the assembled dataset by including the number of images and cells per subset. This would help better understand the training process and the possible impact of imbalances with respect to cell types and imaging modalities.

We have included a table with all dataset details in Appendix A.

2) Extend the comparison with other state-of-the-art (SOTA) methods for the in-domain setting. Given the diverse dataset employed to build CellSAM generalist and specialist models, why not use the actual SOTA tables from the original datasets to compare their performance with other existing approaches when possible? This would help readers to better place CellSAM's performance in context with specialized methods that have been optimized for each dataset.

We added a figures with an overview of all comparisons in S4-S6. We have found that specialist CellPose models to be an adequate surrogate for the SOTA for individual datasets.

3) Expand the comparison with Cellpose using the NeurIPS challenge dataset. To avoid any controversy while comparing CellSAM's performance against Cellpose, it would be both beneficial and very interesting for the target audience to use a dataset for which Cellpose authors have optimized their own method. Moreover, using the NeurIPS challenge dataset (<https://neurips22-cellseg.grand-challenge.org/>) has the added value of importing an extra set of SOTA methods that can be included automatically for comparison. I truly believe the potential readers of the manuscript would appreciate putting the performance of CellSAM in that context, which has very recently become a hot topic in the field.

We thank the reviewer for this suggestion. We performed this experiment and added the results in Figure 2.

4) Improve CellSAM's usability with interactive tools. Enhance CellSAM's usability by integrating it with established bioimage analysis, visualization, and processing tools such as Fiji or Napari. Additionally, consider developing an interactive tool specifically designed for proofreading or few-shot scenarios to streamline user workflows and improve user experience.

We added a Napari plugin to the GitHub repo and included the few-shot training scripts and demo code highlighting how to use features like image tiling.

GitHub: <https://github.com/vanvalenlab/CellSAM>

5) Expand the repository to provide training code. Expand the existing repository to include training scripts for the sake of completeness and reproducibility, enabling users to replicate experiments and further contribute to the project's development.

We thank the reviewer for their comments. We have updated the GitHub repository to improve reproducibility; we are in the process of including training scripts (most were made in the context of performing computational experience and hence are not usable beyond the first authors of this paper). We do note that while this is a “best practice”, in reality, the significant financial costs of training this model will preclude most groups from being able to make use of these scripts. Moreover, there is significant software engineering tooling required for making models of this scale; requiring that this tooling be made open and maintained as software will hinder our group’s ability to make advances, as software maintenance requires significant ongoing investment.

We address all the recommended formatting suggestions offered below.

Minor comments:

* All over the text, please, homogenize the naming of key terms such as “foundation” (sometimes referred to as “foundational”), “generalist” (sometimes referred to as “general”) and “specialist” (sometimes referred to as “specific”).

We synchronized the language used.

* On page 2, the text reads “Cell segmentation is a variant of the instance segmentation problem”, which seems confusing. My understanding is that cell segmentation is just one of the multiple applications of instance segmentation, not a “variant” of it.

This wording was changed. “Cell segmentation is an application of the instance segmentation problem, [...]”

* In Fig. 1, I would clearly specify in the drawing that the ViT is part of the original SAM, as it is done with the decoder.

* On page 6, at the end of the page there are two typos:

** “we observed that CellsAMPreserved its” => “we observed that CellsAM preserved its”.

** “difference in F1 scores was \geq 0.05,” => “difference in F1 scores was $<$ 0.05,”.

Typos fixed.

* On page 7, there is a missing period after “CellsAM enables fast and accurate labeling”.

Typos fixed.

* In Fig. 3, specify what CellsAM model was used in both settings (generalist or specialist). There is also what seems to be an unfinished sentence in the caption: “(b) Application of CellsAM to various biological imaging.”

Typos fixed.

* Revise the text of the references, some of them are missing their respective journals of publication (See for instance [34], [43], [44]...).

Updated references.

* In Appendix A, the size of the dataset images is said to be 512x512 pixels, but SAM’s encoder has an input of 1024x1024. I assume you’re upsampling your images before going through the ViT but, please, confirm that in the text.

* On page 19, you say that you “manually inspected the LIVECell test split to divide the annotation quality into three classes - good, medium, and poor. We randomly selected images in the good split of the validation set for the CellSAM few-shot learning task.”. This is a bit confusing. Did you divide the validation set as well into three classes or did you use the test set as a validation set?

Because we were not doing any training, we just used data from the validation set for our few shot experiments.

* In Fig. S1, there are two typos: “We demonstrated CellSAM-specificand CellSAM-generalsuperior performance” => “We demonstrated CellSAM-specific and CellSAM-general superior performance”. Notices as well “specific” and “general” should be replaced by “specialized” and “generalist”.

Typos fixed.

* In Fig. S1 as well, considered defining the “ZS point”, “ZS Box” and “FT Box” in the caption and using more distinct colors for the bar plots.

Updated the caption to include the definitions.

* In Fig. S3, please specify on which data is the fine-tuning performed. I assume you’re referring to the CellSAM generalist model, so it was trained on all the training partitions of the original dataset. Moreover, if I understand correctly the plot, the results using zero-shot and bounding boxes are better than using fine tuning and the ground truth boxes for the nuclear category. How is that possible?

Fine-tuned refers to the CellSAM-generalist model. It is the SAM model weights fine-tuned on biological images. CellSAM-generalist (fine-tuned box) performance is better (lower error) or at par with the SAM zero-shot performance. These are both done with box prompting.

* In Fig. S4, there are some typos (“CS GD Bbox”) and many undefined acronyms. Also, the colors are not assigned to the same experiments in (a) and (b).

These typos are now fixed.

* The conclusion extracted from Fig. S4a, (“CellSAM can considerably improve its performance with only ten additional fields of views”) seems a bit an overstatement. The improvement seems very marginal in that plot.

We’ve updated the language of the text and removed the adverb.

G. References

The manuscript appropriately acknowledges previous work through accurate and well-placed citations throughout the text.

H. Clarity and context: lucidity of abstract/summary, appropriateness of abstract, introduction and conclusions

The text of the reviewed paper exhibits a high level of clarity, ensuring accessibility for readers within the scientific community. The language employed is precise, technical, and apt for the subject matter, facilitating comprehension of the research methodology and findings.

Additionally, the paper maintains a logical and well-structured flow, guiding readers through the study with ease. To further enhance its quality, attention to defining technical terms, providing contextual information, and offering detailed explanations of the experimental setup would contribute to a more informative and comprehensive paper.

However, it's important to address concerns regarding certain aspects within the manuscript. Specifically, characterizing CellSAM as a foundation model may not align with its training on a specific dataset for a single task, as outlined in Bommasani et al. (2021). Similarly, the assertion that CellSAM unifies biological image analysis workflows lacks substantial support within the manuscript. These points should be carefully reviewed and potentially clarified to ensure the accurate representation of the method's capabilities and contributions.

References

- [1] Greenwald, Noah F., et al. "Whole-cell segmentation of tissue images with human-level performance using large-scale data annotation and deep learning." *Nature biotechnology* 40.4 (2022): 555-565

- [2] Pachitariu, Marius, and Carsen Stringer. "Cellpose 2.0: how to train your own model." *Nature methods* 19.12 (2022): 1634-1641.

- [3] Stringer, Carsen, and Marius Pachitariu. "Cellpose3: one-click image restoration for improved cellular segmentation." *bioRxiv* (2024): 2024-02

- [4] Stringer, Carsen, and Marius Pachitariu. "Transformers do not outperform Cellpose." *bioRxiv* (2024): 2024-04

NMETH-A55681B, "CellSAM: A Foundation Model for Cell Segmentation"

Point-by-point response

Reviewer #1:

Remarks to the Author:

The authors have addressed, or at least attempted to address, most of the comments and suggestions from my previous review, leading to improvements in the overall quality and clarity of the manuscript. However, several important aspects still require further attention.

The authors continue to refer to their proposed method as a “foundation model for cell segmentation,” justifying this on two grounds: (1) that the “underlying SAM model is frequently referred to as a foundation model” and CellSAM builds upon it, and (2) that CellSAM “leverages transformers (both in the ViT and the automated bounding box prompter), self-supervision (inherited from the SAM base), enables multiple applications (accelerated labeling and automated inference via bounding box prompting), and takes advantage of the data/compute scaling laws observed for this architecture.” I find both justifications inadequate.

The first argument implies that any software or method built on top of SAM should be considered a foundation model, which stretches the original definition beyond its intended meaning. Even the authors of SAM themselves questioned whether SAM fits this definition, given that SAM is dedicated solely to segmentation. As for the second argument, while CellSAM indeed leverages transformer architectures and benefits from SAM’s pretrained self-supervised encoder, it does not perform self-supervision on its own and cannot be repurposed for any task other than instance segmentation with bounding box prompts. Although I understand the desire to adopt current terminology, it is important to adhere to established definitions. CellSAM is not a foundation model; rather, it is a method that builds upon SAM through prompt engineering to produce a robust, generalist solution for cell segmentation. This is a noteworthy contribution in its own right, without needing to inflate its classification. A more accurate title for the manuscript could be along the lines of: “CellSAM: A Generalist Cell Segmentation Model Exploiting Foundation Models.”

Per the editor's email this does not need to be addressed

Regarding Section 2.4 ("CellSAM unifies biological image analysis workflows"), the revised manuscript clarifies that a generalist CellSAM model was used across different use cases, which offers the advantage of avoiding the need to train or fine-tune separate models for each scenario. However, the core task remains the same—2D instance segmentation—and what changes is the downstream application in a broader bioimage analysis pipeline (e.g., tracking, 3D segmentation). While these are compelling examples, the section title exaggerates CellSAM's role in unifying workflows. As noted previously, any other generalist cell segmentation method could serve the same purpose. A more accurate title might be: "CellSAM as a Key Enabler of Diverse Bioimage Analysis Workflows."

We will change the title of that section to: "CellSAM Enables Diverse Bioimage Analysis Workflows."

Other Comments:

* I appreciate the inclusion of Appendix C.2, which explains the various metrics. However, the use of a lower IoU threshold (0.3) for human-to-human comparisons compared to human-to-CellSAM comparisons (0.5) seems inconsistent. Wouldn't it be fairer to apply the same threshold in both cases? Additionally, please clarify whether the human-to-CellSAM comparisons were made against annotations from all three annotators or just one.

We thank the reviewer for pointing this out. We did indeed apply the same threshold of 0.3 for both cases. We will correct this in the supplement.

R1) The human-to-CellSAM comparisons were made against all three annotators. CellSAM's predictions were compared to each annotator, and each annotator was compared to every other annotator. We will clarify this in the text.

* The suggested improvements to the few-shot visualization in the LIVECell dataset (adding bounding boxes and using a sparser color map) have not been implemented in Fig. S8, despite the authors' claim to the contrary.

We thank the reviewer for pointing this out and apologize for not updating the plot. We provided 3 samples of the updated version of these plots, which we will add to the final manuscript. We used the same style for masks as in the rest of the manuscript and added

another column for the bounding boxes. We can make additional adjustments if they are requested.

* The inclusion of Fig. S1 showing inference times is appreciated, but further details are needed. Specifically, what models and datasets were used, and on what hardware (e.g., GPU or CPU)? It would also be valuable to include timings for both CPU and GPU scenarios, as many potential users may not have access to a GPU. Moreover, it is curious that Cellpose's execution time does not increase with the number of cells, unlike CellSAM. Any insights into this behavior would be highly informative.

To generate this plot, we randomly sampled two images from each dataset in our test set and filtered them to ensure a diverse range of cell counts. All benchmarks were performed on a single A6000 GPU using CellSAM in non-batched mode. While batched mode offers a 3–5× speedup, all results presented in the paper were obtained using non-batched mode. Batched mode is now available on GitHub for users because we added it as part of our new and ongoing software developments.

Unlike Cellpose, which requires only a single model pass, CellSAM performs one pass for the embedding module and an additional pass for each query (i.e., each cell). As a result, the wall-clock time scales (approximately) linearly with the number of cells.

* The NeurIPS challenge comparison in Fig. 2b is a great addition, providing valuable context for CellSAM's performance in the broader field.

* Appendix D (Hyperparameters) is helpful, but it only lists a final set of hyperparameters. Was this the only set tested, or were other configurations explored? As mentioned previously, following the recommendations by Dodge et al. ("Show your work: Improved reporting of experimental results") would greatly improve transparency by including tables summarizing the hyperparameter search space in the supplementary materials.

We did not perform a systematic full hyperparameter search over all the parameters due to financial limitations. Each experiment was very expensive to run and a comprehensive search was not feasible. Given the cost, we have tried to faithfully report everything useful that we tried in this paper so that future efforts can learn from our experience.

* I was surprised to learn that CellSAM fine-tuning was performed for only one epoch. What was the reasoning behind this decision? Would further training improve performance, particularly for specialist models? Clarifying this would strengthen the paper's methodological rigor.

We will clarify this section in the revised manuscript. The first part of Appendix D refers to the CellFinder training procedure, which involves training both the detection model and the SAM-ViT. This constitutes the bulk of the training. Once this is finished, we need to align the representational interface between the SAM-ViT and the rest of the model. This consists of finetuning CellSAM for a limited number of epochs (50 in the revised version), using a cosine learning rate schedule to mitigate overfitting. At this stage, the SAM-ViT has been exposed to all cellular data and can recognize relevant patterns. The fine-tuning primarily adjusts smaller components of CellSAM—specifically, the "neck" (a feedforward neural network bridging the ViT and the rest of the model)—which is prone to overfitting and thus requires careful tuning.

* There is some inconsistency regarding the value of fine-tuning. Fig. S5 suggests that fine-tuning yields little improvement, as the specialist CellSAM models don't significantly outperform the generalist ones. Conversely, Fig. S7b shows a noticeable improvement in

few-shot settings after fine-tuning. As a potential user, I would appreciate clearer guidance on whether fine-tuning CellSAM is worthwhile for new datasets. Are there differences in the fine-tuning approach used for the results shown in Fig. S5 and Fig. S7b?

S5 refers to Cellpose specialist vs. generalist analysis, S4 is for CellSAM. A better, albeit not directly comparable, is S2, which we believe to be more informative. In S2, the difference between zero-shot (box) and fine-tuned (box) depends on the dataset, which we interpret to be due to the level of difficulty each specific type of cell presents for segmentation. The difference between zero-shot and fine-tuned on Yeast is very small, whereas it is much higher on Tissue. Our general guidance is: if your data is similar to what CellSAM is trained on, fine-tuning is not necessary. If there are significant morphological differences, then fine-tuning is recommended.

* The authors mention in their response that “Training/validation curves are now plotted in Appendix E.” However, this appendix is missing from the revised manuscript.

These are the training curves for the generalist model – these will be added to the revised manuscript

* In Appendix B.1.1, the reported training times lack clarity. Do these numbers apply only to the generalist model, or do they include specialist models as well? Clarifying this would improve the manuscript's transparency.

These numbers are the ones for the CellSAM generalist model. For specialist models, the numbers can just be linearly scaled by the fraction of the total data used (e.g. for 1/9 of the data used for the generalist model, it'll take 1/9 of the time). We will clarify this in the revised manuscript.

* Inference times per full-size image are still missing, which would be highly informative (Fig. S1 only shows inference times as a function of the number of cells). As noted before, please consider providing results for both CPU and GPU scenarios.

We feel that the plots on inference time as a function of number of cells is more accurate, given that our model's speed scales with the number of cells. Even if the FOV is the same size, inference will be longer for more crowded fields of view. We can provide estimates for medium and high levels of crowding so readers can interpret how long inference would take in these circumstances. Given the model size, we feel that most users would use a GPU for inference; we can perform CPU benchmarking at the editor's request.

* I tested the newly added CellSAM Napari plugin but encountered installation issues due to missing libraries. Additionally, I experienced occasional freezing when using manual and automatic annotations, with Napari displaying a "not responding" message. Furthermore, the plugin isn't listed among installable Napari plugins, and there is no accompanying documentation. Proper documentation and smoother installation would significantly enhance usability for the broader bioimage community.

Minor Comments:

* Some terminology inconsistencies remain, such as "generalist" vs. "general" and "specialist" vs. "specific."

We thank the reviewer for pointing this out. We will fix this in the final manuscript.

* Consider including the image sizes (or their range) in pixels in Table 1.

We thank the reviewer for pointing this out. We will fix this in the final manuscript.

* Figures S5 and S6 could be merged for clarity.

We thank the reviewer for the suggestion. We will do this in the final manuscript.

* My previous query about the difference between SAM's encoder input size (1024x1024) and CellSAM's input size (512x512) remains unanswered. Are the images upsampled before being fed into the ViT?

The reviewer is correct - we upsample to 1024x1024.

Remarks on code availability:

I installed the new CellSAM Napari plugin following the instructions in the GitHub repository, but I encountered several issues during the installation process. Rather than installing CellSAM and its Napari plugin directly with pip, I opted for the more commonly recommended approach of creating a virtual environment with conda (Python 3.10). During installation, I discovered that several required libraries were missing, specifically: matplotlib, distributed, dask_image, and sklearn.

Additionally, after installing these dependencies, I had to downgrade numpy from version 2.2 to 2.1 to make the plugin work properly.

Once installed, I experienced occasional freezing while using the plugin, both during manual and automatic annotation, with Napari displaying a "not responding" message.

Furthermore, I was unable to find any documentation associated with the plugin. The plugin also does not appear in the list of installable plugins directly from Napari, which might hinder accessibility for less experienced users. Proper user documentation, as well as a smoother installation process — potentially providing conda installation instructions — would greatly enhance usability for the broader community.

We thank the reviewer for testing this. We will fix these issues and add the Napari plugin as well as a conda setup & pip package.

Additionally, I was unable to access the online demo at <https://cellsam.deepcell.org/>, as it returned the following error:

404: NOT_FOUND

Code: DEPLOYMENT_NOT_FOUND

ID: cdg1::hf7ml-1742389666481-061b2d786098

Ensuring the demo is accessible and functional would be valuable for showcasing the tool's potential to prospective users.

We agree with the reviewer and apologize for the demo being offline. It is now online and running again. Our hosting provider (brev.dev) was acquired by NVIDIA during the review process – we are currently working to port the backend to AWS Lambda which should prevent significant downtime in the future.

Reviewer #2:

None

Reviewer #3:

Remarks to the Author:

Thank the authors for addressing the reviewers' comments and for the substantial revisions to the manuscript. I appreciate the detailed rebuttal letter and the efforts the authors have made to improve CellSAM.

I. Novelty

The authors have argued effectively for the utility and engineering novelty of CellSAM, highlighting the careful selection of AnchorDETR and the overall system design. The authors also make a valid point about the limited algorithmic novelty in cell segmentation in recent years and the importance of high-utility tools. The justification for using the term "foundation model" is also reasonable in the context of leveraging SAM and its practical applications.

While the core algorithmic novelty remains debatable, the revised manuscript and rebuttal adequately address this point by emphasizing utility, engineering effort, and the practical application as a 'foundation model' in the field. I further suggest a proper citation and discussion with recent works like Segment Anything for microscopy [1] and Medical SAM [2].

[1]A. Archit, S. Nair, N. Khalid, P. Hilt, V. Rajashekar, M. Freitag, S. Gupta, A. Dengel, S. Ahmed, and C. Pape, "Segment anything for microscopy," bioRxiv, 2023. [Online].

[2]Ma, J., He, Y., Li, F., Han, L., You, C., & Wang, B. (2024). Segment anything in medical images. *Nature Communications*, 15(1), 654.

We thank the reviewer. We cite [1] in our manuscript ([60]). We add the citation for [2] in the final manuscript and delineate our work with respect to theirs in the discussion. In short, both works use SAM's original workflow to speed up annotation for cells and medical data, label a large dataset, and then fine-tune the original SAM model. We consider both of these papers to be significant contributions to the field. Our work builds on them, enabling automated instance segmentation through an additional detection model that we train on top of the SAM architecture.

II. Data and Experiments

1. Dataset Description:

The authors have added a table in Appendix A detailing the dataset information. This is a significant improvement and addresses the original concern. The table provides valuable information about the datasets used.

2. "Unifying Bioimaging Workflows" Argument:

The authors have clarified that the same CellSAM-generalist model is used for all tasks and argued that its performance justifies this unification. The authors have also revised the text to clarify model usage.

The clarification regarding the use of CellSAM-generalist is helpful. However, while using a generalist model simplifies workflows, it still isn't fully clear what makes CellSAM specifically uniquely positioned to unify workflows compared to other generalist methods

(like Cellpose-generalist, if it existed at a comparable performance level). While the authors argue CellSAM-generalist performs better, further elaborating on the functional advantages of CellSAM in these unified workflows (beyond just generalist segmentation performance) would strengthen this claim. For example, are there specific features of CellSAM's prompting or output format that facilitate easier integration into diverse analysis pipelines compared to other segmentation methods? Further explanation will enhance the significance of the prompting segmentation model.

Given that R1 raised a similar point, we will revise the section title to: “*CellSAM Enables Diverse Bioimage Analysis Workflows.*” We view what the reviewer referred to as “just generalist segmentation performance” not as a minor point, but as a central contribution of this paper. The ability of CellSAM to robustly segment a wide range of cell types with a single model fundamentally simplifies downstream analysis. Researchers no longer need to switch between task-specific models or maintain multiple sets of weights—CellSAM enables a streamlined pipeline with one model download and a single API call that integrates smoothly with various downstream tools. This consolidation significantly reduces complexity, promotes reproducibility, and lowers the barrier to adoption across diverse bioimaging tasks. In this way, CellSAM does not merely perform segmentation—it serves as a drop-in segmentation backend that supports broader ecosystem-level integration. This systemic simplification positions CellSAM as uniquely capable of unifying and accelerating a wide range of bioimaging workflows.

We also note that in addition to accuracy, equivalent or superior performance of generalist models over specialist models is necessary for generalist models to achieve wide adoption; if specialist models perform better, then users will use them as an alternative. CellSAM is the first model (to our knowledge) where the general model performs equal to or better than specialist case. We will amend the discussion to include these points.

3. Insufficient Experimental Comparisons with Cellpose:

The authors have extensively addressed this concern, acknowledging the controversy and providing detailed justification for Cellpose comparison. The authors have also performed new experiments, including training Cellpose models using their recommended procedure, and incorporated comparisons on the NeurIPS dataset. The authors present data suggesting CellSAM outperforms Cellpose even in these comparisons.

The significant effort the authors have put into improving the Cellpose comparisons is commendable. Including the NeurIPS dataset and training Cellpose models using their own protocols significantly strengthens the validity of the performance claims. While debates about benchmarking are ongoing in the field, the authors have made a strong

effort to provide a fair and comprehensive comparison within the scope of this revision. The detailed justification in the rebuttal is also very helpful.

4. Efficiency Analysis (Parameters, Training & Testing Time):

The authors have added inference and training times to the supplementary material (Section B.1 and Fig. S1). Including inference and training times is essential and addresses this concern. This information is crucial for users to assess the practical feasibility of CellSAM.

5. Insufficient Ablation Experiments:

While not explicitly stated in the rebuttals provided, Fig. S3 seems to address the comparison with SAM baseline (zero-shot performance vs. fine-tuned box prompting performance, if I understand correctly).

While Fig. S3 provides some insight, it is still not entirely clear if there is a dedicated ablation study that isolates the effectiveness of "CellFinder" (AnchorDETR box prompting) and "CellSAM" (SAM fine-tuning). Could the authors please clarify in the manuscript (or in a more detailed figure caption for S3) how these components are being evaluated and compared to the SAM baseline? Explicitly stating the contribution of each component (bounding box prompting and fine-tuning) would be beneficial.

The reviewer points correctly out that S3 (updated manuscript) shows the SAM baseline, which is Zero-auto (box). Now, if we had ground truth bounding boxes as prompts, we can improve the segmentation performance, which is Zero-shot (Box). Since ground truth is usually unavailable, we must get them automatically using CellFinder. So the gap from Zero-shot (auto) to Zero-shot (box) is what we get through Cellfinder. If we then fine-tuned other parts of the model (i.e. the neck, explained in the methods), and do the CellSAM/SAM-finetuning we get to the Fine-tuned (Box) bars. This is how the individual parts contribute to overall performance. We will explain this more clearly in the revision.

6. Fine-tuning Comparison (Full vs. Proposed):

Not explicitly addressed in rebuttals provided. While the current fine-tuning approach is effective, a brief comment in the supplementary material or discussion about why full fine-tuning was not pursued (e.g., computational cost, potential overfitting, limited benefit observed in preliminary experiments, or if it was explored and the current approach proved better) would be helpful for completeness.

We thank the reviewer for pointing this out, we will add additional comments to the supplement. In short, we experimented with different training recipes; the one we ended up using was the best tradeoff between computational cost and overfitting. Empirically,

we found that our current recipe mitigated overfitting the best. Due to cost constraints of training each model, this search was guided through experience rather than systematic optimization.

7. Segmentation Performance on Irregular Shapes, Noisy Data:

Not explicitly addressed in rebuttals or manuscript revisions mentioned in rebuttals. This point remains unaddressed. For a model aiming to be a 'foundation model', robustness to noisy data and performance on diverse cell shapes is important. While perfect robustness is unrealistic, a brief discussion in the manuscript acknowledging these limitations and potentially mentioning future directions for improvement (e.g., data augmentation with noisy data, specific architectural adaptations for irregular shapes) would be valuable. The reviewer's comment about it not working well for random samples from their test platform is concerning and should be acknowledged in some way, even if it's difficult to fully quantify without more details on "random samples."

We thank the reviewer for the comment. We agree with the points raised by the reviewer, and we have consistently argued that the best way to ensure model generalization is to expand the support of the training data distribution. While these efforts are ongoing in our lab and others, this current work seeks to answer a different question – given the sum total of labeled training data in the field, what is the best general model that can be made? Performance guarantees only exist for images similar to training data, even for foundation models. We will amend the discussion to clarify this point and avoid an impression that we are overselling what CellSAM can reasonably be expected to do. We note that human-in-the-loop labeling and few-shot fine-tuning are two approaches towards dealing with images that are too far out of distribution, both of which are discussed in the paper.

III. Code Usability

The authors acknowledge the initial lack of polish and claim to have spent time adding polish and new features to improve usability. The repository now includes:

1. Clear installation instructions.
2. Documentation for using the code and pre-trained models.
3. Pre-trained models readily available for download.
4. Example package and website that demonstrate basic usage.

IV. Statistical Analysis

The authors mention reporting mean and standard error for segmentation error in Figure 2. Reporting mean and standard error is a basic but important improvement.

We thank the reviewers for their comments.

The revised manuscript shows significant improvement. Addressing the remaining points outlined above, particularly regarding proper citation of recent works, explanation on the usage of prompting and proposed fine-tuning, clarification of ablation experiments, and discussion of limitations (noisy data, irregular cells), will further strengthen the manuscript and make it more suitable for publication in Nature Methods.

Remarks on code availability:

The authors have provided their codes and models with proper documents. Also, they develop a Napari package for annotation functionality. The utilities can also be accessed online at <https://cellsam.deepcell.org/>. The repository now includes:

1. Clear installation instructions.
2. Documentation for using the code and pre-trained models.
3. Pre-trained models readily available for download.
4. Example package and website that demonstrate basic usage.

I appreciate the renaming of Section 2.4 to “CellSAM Enables Diverse Bioimage Analysis Workflows”, which is more appropriate. However, the caption of Figure 3 still refers to CellSAM as “unifying biological imaging analysis workflows”. For consistency, please update the figure caption as well.

We thank the reviewer and edited the caption.

For Figure S8, consider integrating the bounding boxes and segmentation predictions into the same images using distinct colors for contours and boxes. Enlarging or zooming into the raw images would also improve interpretability.

We thank the reviewer for their suggestion and have edited the figure accordingly. We included the new figures at a very high resolution, so one can easily zoom in on regions of interest without losing image quality.

Thank you for the additional explanation and clarification of inference benchmarking in Figure S1. However, I still believe including CPU-based inference times would significantly benefit readers. As noted, many potential users may not have access to a compatible GPU. Reporting results for both scenarios (CPU and GPU) would provide a more complete picture of practical usability.

We thank the reviewer for their suggestion. We added another plot (see below) to the supplementary materials, comparing the average per-image-inference times for CellSAM and Cellpose on GPU vs. CPU.

Table 1 still does not include image sizes (or pixel size ranges), which would help contextualize the diversity of the datasets used.

We thank the reviewer for their suggestion and have included image sizes across all of our datasets. With regard to pixel sizes, this is challenging as several of the datasets we use do not report consistent pixel sizes on a per image basis, or are themselves sourced from

multiple locations (e.g., the Cellpose dataset). Where possible, we have reported pixel sizes either from the dataset, from the camera indicated in the paper, or from an estimation based on the cell type present in the dataset.

Figures S5 and S6 have not been merged, and their captions continue to use the term “specific” instead of “specialist”. Consistent terminology would improve clarity.

We thank the reviewer for their suggestion and edited the plot accordingly (see below).

One small but important recommendation is to create an official release of the code (e.g., via GitHub Releases or Zenodo). This would allow the codebase to be formally linked to the publication and would also facilitate reproducibility and version tracking for users referencing the method in the future.

Finally, I note that the Napari plugin is still not listed in the official Napari hub. Additionally, a plugin named “samcell-napari” does appear, which could easily be confused with CellSAM. I recommend addressing this potential ambiguity—either by

clarifying the naming or ensuring the correct plugin is discoverable in the hub—to avoid confusion for users.

The points I raised previously regarding installation and documentation of the Napari plugin remain either unaddressed or pending implementation. As of testing the current version, installation required multiple manual dependency fixes, and the plugin still lacks user documentation and is not listed in the official Napari plugin registry. These issues may hinder adoption and should be resolved before public release.

We thank the reviewer for this comment. The concerns regarding installation should now be fully addressed by the [napari plugin documentation page](<https://vanvalenlab.github.io/cellSAM/napari.html>) which is included as a top-level page in the official documentation. The documentation includes installation recipes that will work for users regardless of whether they have downloaded the source code. Additionally, the documentation includes a demonstration of how to use the napari plugin which should address usability concerns.

Re: napari-hub - the authors have chosen explicitly not to publish the plugin on napari hub for two main reasons. The first is related to best practices for software distribution. The library itself and the napari plugin are still under active development, so stable releases have not yet been made. This is intentional until the software becomes stable enough and regression testing is extensive enough that the authors can make reasonable backward-compatibility guarantees. While publishing to a centralized package hosting platform like napari hub can indeed increase discoverability, premature publication of "stable" software can negatively impact user experience when backward compatibility guarantees are not in place. Furthermore, while CellSAM is capable of interactive segmentation on a per-cell basis, the true power and innovation of the model derive from the full detection+segmentation capabilities; i.e., the combined capabilities of the CellFinder and instance segmentation components. The authors strongly encourage users to consider the full instance segmentation workflow as opposed to one that relies on interactive instance segmentation. To this end, the documentation has placed more emphasis on such use-cases; e.g. with the [example gallery](https://vanvalenlab.github.io/cellSAM/auto_examples/index.html). Napari is still used as part of this workflow, but via library interface to visualize results of the full instance segmentation pipeline, rather than a GUI-centric, interactive segmentation approach. This workflow better captures the recommended usage pattern and distinguishing features of the CellSAM model.

Finally, re: naming collision on napari hub - this is an unfortunate consequence of the open nature of the hub hosting structure. Unlike popular packaging indices like PyPI, there does not seem to exist a mechanism for resolving naming issues (cf. PEP 541). The authors hope that the [official documentation](<https://vanvalenlab.github.io/cellSAM/index.html>) is sufficient as a single, centralized point of reference for the project, including the napari plugin.